# Baseline eGFR cutoff for increased risk of post-contrast acute kidney injury in patients undergoing percutaneous coronary intervention for ST-elevation myocardial infarction in the emergency department

**Je Sung You[1,2], Junho Cho[1], Hye Jung Shin[3], Jin Ho Beom[1]***

**1** Department of Emergency Medicine, Yonsei University College of Medicine, Seoul, Republic of Korea,
**2** Department of Emergency Medicine, Yonsei University College of Medicine, Gangnam Severance Hospital, Seoul, Republic of Korea, **3** Department of Biomedical Systems Informatics, Biostatistics Collaboration Unit, Yonsei University College of Medicine, Seoul, Republic of Korea

* wangtiger@yuhs.ac

**Data Availability Statement:** All relevant data are within the paper and its Supporting Information files.

## Abstract

Acute myocardial infarction is an acute-stage disease that requires prompt diagnosis and treatment. Primary percutaneous coronary intervention (pPCI) for ST-elevation myocardial infarction (STEMI) is a high-risk factor for post-contrast acute kidney injury (PC-AKI). This retrospective cohort study analyzed the data of 754 patients with STEMI who underwent pPCI and were integrated into the Fast Interrogation Rule for STEMI critical pathway program between 2015 and 2019. We aimed to determine the optimal cutoff baseline eGFR for identifying a high risk of PC-AKI after multivariable adjustment with statistically significant risk factors. We also compared the incidence rates of PC-AKI between the previous and current diagnostic criteria. The probability of PC-AKI increased when the baseline estimated glomerular filtration rate (eGFR) was $\leq$ 79mL/min/1.73 m$^2$. The optimal cutoff baseline eGFR for high risk of PC-AKI was found to be an eGFR of $\leq$ 61 mL/min/1.73 m$^2$ after multivariable adjustment. The current diagnostic criteria more accurately identified the patient group with impaired renal function. Our results have clinically significant implications for identifying patients at a high risk of developing PC-AKI, especially before and after the use of contrast agents in patients who require PCI for STEMI in the emergency department.

## Introduction

Acute myocardial infarction (AMI) is one of the acute-stage diseases that requires prompt diagnosis and treatment in the emergency department (ED) and is associated with a high risk of mortality [1,2]. In particular, we can expect favorable outcomes only if primary percutaneous coronary intervention (pPCI) is conducted early in patents with ST-segment elevation myocardial infarction (STEMI) on electrocardiography [3,4]. However, acute coronary

**Funding:** Funding: This study was supported by a grant from the National Research Foundation of Korea (NRF) grant funded by the Ministry of Science, ICT & Future Planning (NRF - 2019R1C1C1006332 to J.H.B., and NRF - 2021R1C1C1009209 to J.S.Y.) and a faculty research grant from Yonsei University College of Medicine for 6-2020-0086. The funders had no role in study design, data collection and analysis, decision to publish, or preparation of the manuscript.

**Competing interests:** NO authors have competing interests Enter: The authors have declared that no competing interests exist.

syndrome (ACS) is a high-risk state due to highly thrombogenic state, increased inflammation, and decreased renal perfusion through vasoconstriction or hemodynamic instability. exposure of contrast medium can increase chances of post-contrast acute kidney injury (PC-AKI) [5–7]. Many studies have reported that PC-AKI aggravates clinical outcomes, with repeated revascularization, extended length of hospital stay, increased costs, and high rates of short- and long-term mortality [8–10]. PC-AKI can cause irreversible kidney damage and even death in severe cases. Unlike computed tomography (CT) where the contrast medium is injected intravenously, for PCI, the contrast is directly injected into the artery, and this is considered to be associated with the incidence of PC-AKI [11,12]. The incidence of PC-AKI is high in patients undergoing cardiac procedures such as PCI [6,13].

Intravascular volume expansion can effectively prevent post-contrast acute kidney injury (PC-AKI) following invasive angiography. Numerous studies have been conducted to prevent PC-AKI, employing various strategies such as appropriate risk stratification, reducing iodinated contrast dose, utilizing low or iso-osmolar contrast media, withholding nephrotoxic medications when necessary, and administering protective agents [14,15]. However, the use of contrast medium is associated with many adverse effects; hence, it is a burden on physicians in the clinic setting [7]. Therefore, to prevent PC-AKI, it is paramount to recognize risk factors in emergent situations or in the perioperative period [16,17]. Studies have shown that certain patient groups with baseline renal function, such as those with estimated glomerular filtration rates (eGFRs) <60, are at an increased risk of developing PC-AKI. and the 2018 ESUR guidelines suggested that the eGFR level for PC-AKI risk to be <45 mL/min/1.73 m$^2$ without considering various risk factors; however, few studies aimed to identify baseline eGFR cutoff values for predicting the occurrence of PC-AKI. As many patients with critical illness have various risk factors, it is difficult to simply predict the occurrence of AKI with baseline values of renal function tests performed immediately upon admission to the ED.

Therefore, we aimed to determine the cutoff baseline renal function that can be used to identify the risk of PC-AKI in patients who undergo PCI for STEMI in the ED along with various risk factors in regard to the findings of previous studies. Using the probabilities of PC-AKI and rates of survival to discharge according to the baseline renal function, we compared the differences between the recently updated diagnostic criteria and previous diagnostic criteria and provided clinical evidence for the new diagnostic criteria.

## Materials and methods

### Study population and Fast Interrogation Rule for ST-elevation MI (FIRST)

This retrospective, observational cohort study used a registry of a critical pathway in a single tertiary academic hospital that attends to 100,000 patients in the ED annually. This study was conducted according to the guidelines of the Declaration of Helsinki and was reviewed and approved by the institutional review board of Yonsei University Health System (No. 4-2020-1256). The need for written informed consent was waived owing to the retrospective study design.

To provide proper management to patients with STEMI, a multidisciplinary critical pathway based on a computerized provider order entry (CPOE) system, known as the Fast Interrogation Rule for ST-elevation MI (FIRST), was implemented in our institution in 2007. This critical pathway was made to provide standard treatment and reduce unnecessary in-hospital time delays through a CPOE-based alert system, short message service, and simple standing orders in patients with STEMI [18]. The critical pathway was activated according to electrocardiogram (ECG) criteria and typical clinical features of chest pain following the standard STEMI guidelines.

When a patient met the criteria for ST-segment elevation on ECG with typical complaints of chest pain on ED arrival, the ED physician activated the FIRST critical pathway by clicking

the activation button on the order entry window [18]. Once activated, the on-call cardiologist was consulted. The on-call cardiologist immediately evaluated the patient and applied standard treatment following the guidelines of the American College of Cardiology /American Heart Association [19].

This study included consecutive patients who were activated in the FIRST critical pathway between January 1, 2015, and December 31, 2019. We collected data for this study from December 26, 2020, to December 25, 2021, and analyzed the data. We analyzed patients admitted for STEMI who underwent primary PCI. However, we excluded patients with Do Not Attempt Resuscitation status, brain hemorrhage, active gastrointestinal bleeding, pericarditis, history of coronary artery bypass graft surgery, arrhythmia, pericardial effusion, stress-induced cardiomyopathy, pulmonary embolism, STEMI judged to be unclear by the cardiologist, and those who refused to undergo PCI. We did not have access to any information that could identify individual participants during or after data collection.

Fig 1 presents the final enrollment and exclusion criteria along with the occurrence of PC-AKI in patients with STEMI who underwent PCI.

## Primary percutaneous coronary intervention and definition of PC-AKI

Unless contraindicated in the ED, we treated all patients with a loading dose of 600 mg of clopidogrel and 300 mg of acetylsalicylic acid, and the on-call intervention team performed pPCI according to the FIRST protocol, based on standardized treatment. According to the new 2018 European Society of Urogenital Radiology (ESUR) guidelines, our hospital has implemented fluid therapy to prevent the occurrence of PC-AKI at our practice site. The IV infusion rate was also adjusted with fluids based on the guideline's recommendations for normal saline (0.9% sodium chloride) at a rate of 1ml/kg/hr, except in patients with severe heart failure or left ventricular (LV) dysfunction.

The average time between admission to the ED and receiving PCI was 68 (median, [interquartile range, 54, 89]) minutes. For coronary angiography, the following contrast mediums were used: low-osmolar, monomeric iopamidol (370 mg iodine/mL, 796 mOsm/kg of water; Scanlux, Sanochemia Pharmazeutika, Austria; for 59.42% of all patients); nonionic, iso-osmolar iodixanol (320 mg iodine/mL, 290 mOsm/kg of water; Visipaque, GE Healthcare Inc., Princeton, NJ, USA; for 39.25% of all patients); and nonionic, low-osmolar, monomeric iobitridol (300 mg iodine/mL, 915 mOsm/kg of water; XenetiX, Guerbet, Aulnay-Sous-Bois, France; for 1.33% of all patients). The interventional cardiologist also determined the access sites for PCI, pharmacological and reperfusion therapy, and application of an intra-aortic balloon pump.

In the present study, we calculated the eGFR using the Chronic Kidney Disease Epidemiology Collaboration equation (https://www.kidney.org/content/ckd-epi-creatinine-equation-2009). We divided the enrolled patients into four groups according to the baseline eGFR (mL/min/1.73 m$^2$) values that indicated increased risk of PC-AKI as follows: eGFR < 30, eGFR 30–59, eGFR 60–89, and eGFR ≥ 90. Although the current definition of PC-AKI differs in each guideline, we applied the one of absolute increase in serum creatinine (sCr) levels by > 0.3 mg/dL or a relative increase in sCr levels by > 50% from the baseline within 72 hours after exposure to the contrast medium [20].

## Comparison of previous and current PC-AKI diagnostic criteria

The previous criteria defined PC-AKI as an increase in serum creatinine (SCr) levels by ≥ 0.5 mg/dL or ≥ 25% from the baseline within 72 hours after contrast radiography using an iodinated contrast medium [21]. In this study, we applied both the previous and current definitions of PC-AKI to identify the actual probability of occurrence according to the baseline eGFR and to decide whether there is a statistically significant difference between each baseline eGFR

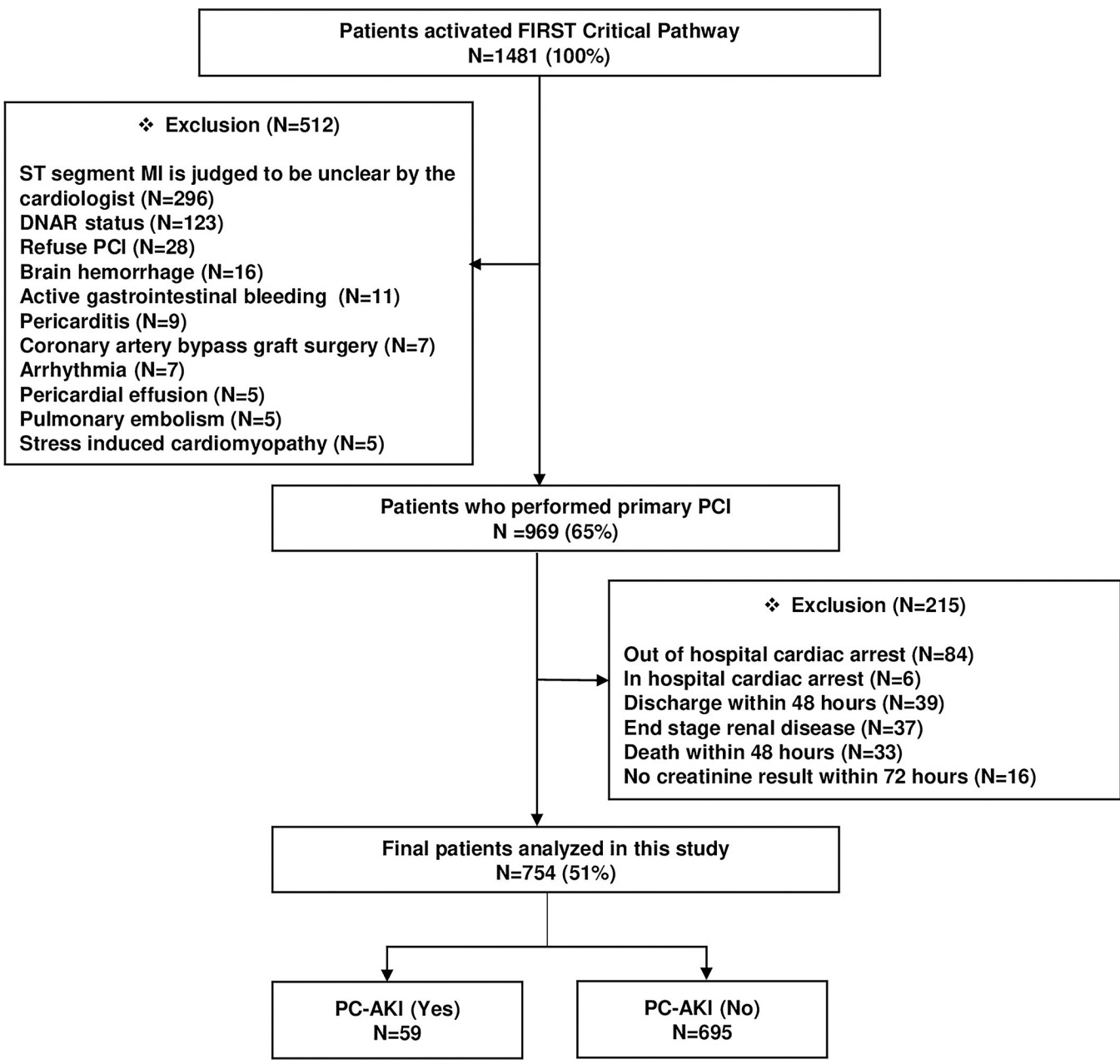

**Fig 1. Flow diagram of enrolled patients.** FIRST, Fast Interrogation Rule for ST-elevation MI; MI, myocardial infarction; DNAR, Do Not Attempt Resuscitation; PCI, percutaneous coronary intervention; PC-AKI, post contrast- acute kidney injury.

group according to the two criteria. Furthermore, we used a mediation analysis to examine the effect of PC-AKI on the rate of survival to discharge by applying the cut-off point baseline eGFR for PC-AKI obtained in this study and dividing it according to the two criteria.

## Data collection

We collected the data on patient demographics, previous medical history, use of nephrotoxic drug type and volume of contrast medium, Killip class, pre-procedural status, left ventricular ejection fraction (LVEF), laboratory test results, baseline eGFR, door-to-balloon time, and PCI

results. We also measured the sCr concentration of all patients at ED admission and daily for 72 hours after exposure to the contrast medium. Considering the definition of PC-AKI, we calculated the changes in SCr and checked the development of PC-AKI.

### Clinical outcomes

According to the baseline renal function, the primary endpoint was PC-AKI incidence in patients who underwent PCI with STEMI. In addition, we derived the optimal cutoff baseline eGFR for identifying the high risk of PC-AKI after multivariable adjustment with statistically significant risk factors as the secondary endpoint. Lastly, we also analyzed the rate of survival to discharge according to the incidence of AKI in patients who underwent PCI.

### Statistical analyses

Demographic and clinical data were presented as median (interquartile range), mean and standard deviation, percentage, or frequency, as appropriate. We compared continuous variables using two-sample t tests or Mann–Whitney U tests, and we also compared categorical variables using chi-squared or Fisher's exact tests. To find the incidence of PC-AKI according to the baseline eGFR groups, we conducted one-way analyses of variance and also performed Bonferroni post-hoc tests for comparisons among multiple groups. A univariable analysis was performed to assess the relationships among the demographic characteristics and clinical data. To identify the independent predictors of PC-AKI and to confirm the relationship between PC-AKI occurrence and survival to discharge using odds ratios, the variables with $p < 0.05$ in the univariable analysis were input as major covariates in the multivariable logistic regression analysis model. We presented results as odds ratios (ORs) and 95% confidential intervals (CIs). To identify the ability of the baseline eGFR to predict the development of PC-AKI, we also plotted receiver-operating characteristic (ROC) curves and determined the optimal cutoff baseline eGFR using the maximum area under the curve (AUC) and Youden's index. A mediation analysis was carried out using the medflex package. A mediation analysis quantifies the degree to which a variable participates in the transmittance of change from a cause to its effect.

  We conducted statistical analyses using SAS, version 9.4 (SAS Institute Inc., Cary, NC, USA) and MedCalc Statistical Software, version 16.4.3 (MedCalc Software bvba, Ostend, Belgium). Differences with $p < 0.05$ were considered significant.

## Results

Fig 1 presents the enrollment flow, exclusion criteria, and clinical outcomes of patients with STEMI who developed PC-AKI. In all, 969 patients underwent primary PCI; of these, 215 patients were excluded because of out-of-hospital cardiac arrest (n = 84), in-hospital cardiac arrest (n = 6), end-stage renal disease (n = 37), discharge (n = 39), death (n = 33) within 48 hours, and absence of creatinine results within 72 hours (n = 16). Finally, 754 patients were included in the analysis. PC-AKI occurred in 59 (7.8%) patients (Fig 1): in 7 (11.9%) patients with an eGFR of $< 30$ mL/min/1.73 m$^2$, in 34 (57.6%) patients with eGFR of 30–59 mL/min/1.73 m$^2$, in 14 (23.7%) patients with eGFR of 60–89 mL/min/1.73 m$^2$, and in 4 (6.8%) patients with eGFR of $\geq 90$ mL/min/1.73 m$^2$.

  The clinical characteristics of the patients in this study were analyzed as follows (Table 1).

  The incidences of PC-AKI significantly differed among the baseline eGFR groups ($p < .001$) (Table 2A). In the post-hoc analysis, we found that the group with an eGFR of $< 60$ mL/min/1.73 m$^2$ had a higher incidence of PC-AKI than the other groups, and there were no significant differences among the remaining groups (Table 2B).

**Table 1. Patient clinical characteristics.**

| Variables | Total n = 754 (100%) | No PC-AKI n = 695 (92.2%) | PC-AKI n = 59 (7.8%) | p-value |
|---|---|---|---|---|
| **Baseline eGFR** median (IQR) | 90 (71, 103) | 92 (75, 103) | 49 (33, 68) | < .001 |
| **Baseline eGFR group** (mL/min/1.73 m$^2$), n (%) | | | | < .001 |
| < 30 | 15 (2.0%) | 8 (1.2%) | 7 (11.9%) | |
| 30–59 | 101 (13.4%) | 67 (9.6%) | 34 (57.6%) | |
| 60–89 | 256 (34.0%) | 242 (34.8%) | 14 (23.7%) | |
| ≥ 90 | 382 (50.7%) | 378 (54.4%) | 4 (6.8%) | |
| **Year,** n (%) | | | | 0.084 |
| 2015 | 134 (17.8%) | 122 (91.0%) | 12 (9.0%) | |
| 2016 | 149 (19.8%) | 130 (87.3%) | 19 (12.8%) | |
| 2017 | 176 (23.3%) | 166 (94.3%) | 10 (5.7%) | |
| 2018 | 150 (19.9%) | 139 (92.7%) | 11 (7.3%) | |
| 2019 | 145 (19.2%) | 138 (95.2%) | 7 (4.8%) | |
| **Age (years)** median (IQR) | 64 (54, 74) | 63 (53, 73) | 75 (65, 79) | < .001 |
| ≥ 65, n (%) | 366 (48.5%) | 321 (46.2%) | 45 (76.3%) | < .001 |
| ≥ 75, n (%) | 181 (24.0%) | 150 (21.6%) | 31 (52.5%) | < .001 |
| **Sex, male,** n (%) | 581 (77.1%) | 541 (77.8%) | 40 (67.8%) | 0.078 |
| **Previous medical history,** n (%) | | | | |
| Hypertension | 416 (55.2%) | 370 (53.2%) | 46 (78.0%) | 0.001 |
| Diabetes mellitus | 200 (26.5%) | 170 (24.5%) | 30 (50.9%) | < .001 |
| Hypercholesterolemia | 67 (8.9%) | 62 (8.9%) | 5 (8.5%) | 0.908 |
| CAD | 102 (13.5%) | 91 (13.1%) | 11 (18.6%) | 0.231 |
| Heart failure | 9 (1.2%) | 5 (0.7%) | 4 (6.8%) | 0.003 |
| Arrhythmia | 20 (2.7%) | 19 (2.7%) | 1 (1.7%) | >.999 |
| Stroke | 30 (4.0%) | 27 (3.9%) | 3 (5.1%) | 0.724 |
| Malignancy | 55 (7.3%) | 49 (7.1%) | 6 (10.2%) | 0.429 |
| Liver disease | 5 (0.7%) | 4 (0.6%) | 1 (1.7%) | 0.335 |
| CKD | 21 (2.8%) | 11 (1.6%) | 10 (17.0%) | < .001 |
| LV EF (%) | 48 (40, 55) | 48 (40, 55) | 36 (28, 44) | < .001 |
| **Contrast amount (ml)** median (IQR) | 190 (150, 250) | 190 (150, 250) | 190 (150, 240) | 0.954 |
| **Type of contrast medium,** n (%) | | | | |
| Scanulx | 268 (59.4%) | 249 (59.6%) | 19 (57.6%) | 0.746 |
| Visipaque | 177 (39.3%) | 163 (39.0%) | 14 (42.4%) | |
| Xenetix | 6 (1.3%) | 6 (1.4%) | 0 (0%) | |
| **PCI results,** n (%) | | | | |
| CAOD 1VD | 280 (37.2%) | 263 (38.0%) | 17 (28.8%) | 0.02 |
| CAOD 2VD | 207 (27.5%) | 194 (28.0%) | 13 (22.0%) | |
| CAOD 3VD | 202 (26.9%) | 176 (25.4%) | 26 (44.1%) | |
| Normal & minimal CAOD | 63 (8.4%) | 60 (8.7%) | 3 (5.1%) | |
| **Pre-procedural status** | , n (%) | | | |
| IABP | 23 (9.2%) | 15 (8.2%) | 8 (11.9%) | < .001 |
| Cardiogenic shock | 132 (52.6%) | 102 (55.4%) | 30 (44.8%) | |
| Heart failure | 96 (38.3%) | 67 (36.4%) | 29 (43.3%) | |
| **Killip class,** n (%) | | | | |

*(Continued)*

**Table 1.** (Continued)

| Variables | Total n = 754 (100%) | No PC-AKI n = 695 (92.2%) | PC-AKI n = 59 (7.8%) | p-value |
|---|---|---|---|---|
| I | 567 (75.6%) | 547 (79.2%) | 20 (33.9%) | < .001 |
| II | 57 (7.6%) | 52 (7.5%) | 5 (8.5%) | |
| III | 48 (6.4%) | 33 (4.8%) | 15 (25.4%) | |
| IV | 78 (10.4%) | 59 (8.5%) | 19 (32.2%) | |
| **Door to balloon time (mins),** median (IQR) | 68 (54, 89) | 66 (53, 86) | 85 (65, 135) | < .001 |
| **Laboratory data,** median (IQR) | | | | |
| WBC ($10^3$/μL) | 9.85 (7.98, 12.76) | 9.83 (7.95, 12.75) | 9.91 (8.17, 14.1) | 0.659 |
| Hemoglobin (g/dL) | 14.4 (13, 15.6) | 14.6 (13.2, 15.6) | 13.2 (10.4, 14.3) | < .001 |
| Hematocrit (%) | 42.7 (38.9, 45.9) | 43 (39.3, 46) | 39 (32.8, 43) | < .001 |
| Delta neutrophil index (%) | 0 (0, 0.7) | 0 (0, 0.6) | 0.3 (0, 1.1) | 0.016 |
| Platelet ($10^3$/μL) | 234 (190, 283) | 235 (192, 283) | 215 (158, 277) | 0.046 |
| Neutrophil (%) | 68.3 (52.9, 79.6) | 68 (52.7, 79.3) | 74.7 (55.4, 83.1) | 0.076 |
| Lymphocyte (%) | 22.6 (13.4, 35.5) | 23 (13.7, 35.7) | 17.8 (10.2, 33.7) | 0.042 |
| Glucose (mg/dL) | 158 (130, 209) | 155.5 (129, 203.5) | 175 (141, 264) | 0.017 |
| BUN (mg/dL) | 17.2 (14.2, 21.6) | 16.9 (13.9, 21) | 25.5 (18.3, 31) | < .001 |
| Uric acid (mg/dL) | 5.6 (4.5, 6.7) | 5.6 (4.5, 6.7) | 6.15 (4.6, 7.5) | 0.052 |
| Total cholesterol (mg/dL) | 182 (151, 217) | 184 (153, 220) | 157 (139.5, 194.5) | 0.002 |
| Creatinine kinase (U/L) | 137 (89, 291) | 134 (88, 280) | 197 (98, 357) | 0.063 |
| CK-MB (ng/mL) | 4.15 (2, 16.35) | 4 (2, 14.3) | 5.8 (3.3, 24.4) | 0.003 |
| Troponin T (pg/mL) | 40 (12, 248) | 36 (11, 223) | 131 (36, 528) | < .001 |
| NT pro BNP (pg/mL) | 216 (53, 1267) | 178 (49, 1025) | 3632 (524, 10479) | < .001 |
| Triglyceride (mg/dL) | 99 (68, 147) | 99 (68, 149) | 96 (65, 119) | 0.218 |
| HDL-Cholesterol (mg/dL) | 40 (34, 47) | 40 (34, 47) | 38 (30, 46) | 0.056 |
| LDL-Cholesterol (mg/dL) | 98 (70, 121.5) | 99 (72, 122) | 79 (50, 98) | < .001 |
| CRP (mg/L) | 5.95 (2, 16.95) | 5.8 (2, 15.55) | 11.55 (2.15, 42.85) | 0.048 |
| **Use of nephrotoxic medication,** n (%) | | | | |
| Before[a] ACEI/ARB | 95 (13.0%) | 84 (12.5%) | 11 (20%) | 0.110 |
| Before Beta blocker | 50 (6.9%) | 45 (6.7%) | 5 (9.1%) | 0.415 |
| Before Statin | 143 (19.6%) | 129 (19.1%) | 14 (25.5%) | 0.257 |
| Before Insulin | 6 (0.8%) | 3 (0.5%) | 3 (5.5%) | 0.007 |
| Before oral DM | 130 (17.8%) | 112 (16.6%) | 18 (32.7%) | 0.003 |
| Before NSAID | 8 (1.1%) | 8 (1.2%) | 0 (0%) | >.999 |
| After[b] ACEI /ARB | 467 (61.9%) | 454 (65.3%) | 13 (22.0%) | < .001 |
| After Beta blocker | 140 (18.6%) | 130 (18.7%) | 10 (17.0%) | 0.739 |
| After Statin | 717 (95.1%) | 664 (95.5%) | 53 (89.8%) | 0.061 |
| After Insulin | 95 (12.6%) | 72 (10.4%) | 23 (39.0%) | < .001 |
| After oral DM | 95 (12.6%) | 90 (13.0%) | 5 (8.5%) | 0.320 |
| After NSAID | 11 (1.5%) | 11 (1.6%) | 0 (0%) | >.999 |
| ECMO, n (%) | 23 (3.1%) | 14 (2.0%) | 9 (15.3%) | < .001 |
| Mehran 2 risk score median (IQR) | 9 (8,11) | 9 (8,10) | 12 (11,14) | < .001 |
| Survival to discharge n (%) | 720 (95.5%) | 678 (97.6%) | 42 (71.2%) | < .001 |

(*Continued*)

**Table 1.** (Continued)

| Variables | Total n = 754 (100%) | No PC-AKI n = 695 (92.2%) | PC-AKI n = 59 (7.8%) | p-value |
|---|---|---|---|---|
| Survival at 30 days n (%) | 725 (96.2%) | 682 (98.1%) | 43 (72.9%) | < .001 |

Data are presented as median (interquartile range), for continuous variables, number (%) for categorical variables. PC-AKI, post contrast- acute kidney injury; eGFR, estimated glomerular filtration rate; IQR, interquartile range; CAD, coronary artery disease; CKD, chronic kidney disease; LV EF, left ventricular ejection fraction; CAOD, coronary artery obstructive disease; VD, vessel disease; IABP, intra-aortic balloon pump; W.B.C., White blood cell; BUN: blood urea nitrogen; CK, Creatinine kinase; NT pro-BNP, N-terminal pro-B-type natriuretic peptide; HDL, high density lipoprotein; LDL, low density lipoprotein; CRP,C-reactive protein; ACEI, angiotensin-converting enzyme inhibitor; ARB, angiotensin receptor blocker; DM, Diabetes mellitus; NSAID, nonsteroidal anti-inflammatory drug; ECMO, extracorporeal membrane oxygenation; Before[a]: Medication history before pPCI; After[b]: Medication history within 3 days after pPCI.

In our study, we identified several variables that affected the incidence of PC-AKI after PCI in patients with STEMI, namely eGFR, age and previous medical history of hypertension, diabetes mellitus, heart failure, chronic kidney disease, and LVEF (%). Additionally, we investigated intra-aortic balloon pumping status, cardiogenic shock and heart failure as influencing variables of the pre-procedural status.

In laboratory data, the following variables were found to have an effect on PC-AKI. Hemoglobin, hematocrit, glucose, BUN, uric acid, total cholesterol, NT pro BNP, and LDL-cholesterol. We also identified the effects of certain nephrotoxic agents, such as insulin level before contrast injection, use of oral diabetes medications, use of angiotensin-converting enzyme inhibitors/ angiotensin receptor blockers (ACEIs/ARBs) after exposure to the contrast medium, and use of insulin medication, on the occurrence of PC-AKI.

As per the multivariable model, the occurrence of PC-AKI in patients who underwent PCI was attenuated by an increased baseline eGFR before the procedure (OR, 0.941; 95% CI, 0.907,0.976; p = 0.001). However, the occurrence of PC-AKI was significantly influenced by heart failure status and development of cardiogenic shock before the procedure (OR,6.747; 95% CI, 1.225,37.161; p = 0.028), (OR, 5.906; 95% CI, 1.363, 25.591; p = 0.018), respectively (Table 3).

We obtained the maximum AUCs in the ROC curve analysis for the variables and found that the baseline eGFR value corresponding to the maximum AUC value was eGFR $\leq$79 mL/min/1.73 m$^2$ (see Fig 2A). Furthermore, the optimal cutoff value baseline eGFR yielded by the multivariable analysis was $\leq$61 mL/min/1.73 m$^2$ (see Fig 2B).

**Table 2. A. PC-AKI incidence based on baseline eGFR.** B. Post-hoc analysis for comparison of PC-AKI incidence among baseline eGFR groups.

| Group | Baseline eGFR (mL/min/1.73 m$^2$) | PC-AKI No (N = 695) n (%) | PC-AKI Yes (N = 59) n(%) | p-value |
|---|---|---|---|---|
| 1 | < 30 | 8 (1.2%) | 7 (11.9%) | < .001 |
| 2 | 30–59 | 67 (9.6%) | 34 (57.6%) | |
| 3 | 60–89 | 242 (34.8%) | 14 (23.7%) | |
| 4 | $\geq$ 90 | 378 (54.4%) | 4 (6.8%) | |

| Adjusted p* | | | | | |
|---|---|---|---|---|---|
| 1 vs. 2 | 1 vs. 3 | 1 vs. 4 | 2 vs. 3 | 2 vs. 4 | 3 vs. 4 |
| >.999 | < .001 | < .001 | < .001 | < .001 | 0.06 |

PC-AKI, post-contrast acute kidney injury; eGFR, estimated glomerular filtration rate.

**Table 3. Risk factors of PC-AKI.**

| Variable | Univariable analysis | | Multivariable analysis | |
|---|---|---|---|---|
| | Odds Ratio (95% CI) | p-value | Odds Ratio (95% CI) | p-value |
| **Baseline eGFR** | 0.938 (0.925, 0.951) | < .001 | 0.941 (0.907, 0.976) | 0.001 |
| **Age (years)** | 1.062 (1.037, 1.087) | < .001 | 0.992 (0.934, 1.053) | 0.788 |
| **Sex, male** | 0.599(0.337, 1.065) | 0.081 | | |
| **Previous medical history** | | | | |
| Hypertension | 3.108(1.650, 5.855) | 0.001 | 1.264 (0.376, 4.248) | 0.705 |
| Diabetes mellitus | 3.195(1.864, 5.477) | < .001 | 0.472 (0.053, 4.195) | 0.501 |
| Hypercholesterolemia | 0.945(0.365, 2.451) | 0.908 | | |
| CAD | 1.521(0.762, 3.036) | 0.234 | | |
| Heart failure | 10.036(2.62, 38.451) | 0.001 | 2.165 (0.202, 23.198) | 0.523 |
| Arrhythmia | 0.613(0.081, 4.665) | 0.637 | | |
| Stroke | 1.326(0.390, 4.506) | 0.651 | | |
| Malignancy | 1.492(0.611, 3.645) | 0.379 | | |
| Liver disease | 2.978(0.328, 27.086) | 0.333 | | |
| CKD | 12.691(5.139,31.342) | < .001 | 2.123(0.665, 6.785) | 0.204 |
| LVEF (%) | 0.926(0.905, 0.947) | < .001 | 0.971 (0.930, 1.014) | 0.187 |
| **Contrast amount (cc)** | 1.000(0.995, 1.004) | 0.859 | | |
| **Pre-procedural status** | | | | |
| NA | ref | - | | |
| IABP | 15.635 (5.913, 41.339) | < .001 | 2.688 (0.256, 28.265) | 0.410 |
| Cardiogenic shock | 7.247 (3.770, 13.928) | < .001 | 5.906 (1.363, 25.591) | 0.018 |
| Heart failure | 11.275(4.767, 26.667) | < .001 | 6.747 (1.225, 37.161) | 0.028 |
| **Laboratory data** | | | | |
| WBC ($10^3$/μL) | 1.042 (0.979, 1.110) | 0.196 | | |
| Hemoglobin (g/dL) | 0.711 (0.631, 0.800) | < .001 | 1.012 (0.739, 1.384) | 0.942 |
| Hematocrit (%) | 0.893 (0.856, 0.931) | < .001 | | |
| Delta neutrophil index (%) | 1.050 (0.917, 1.203) | 0.482 | | |
| Platelet ($10^3$/μL) | 0.997 (0.993, 1.000) | 0.079 | | |
| Neutrophil (%) | 1.012 (0.996, 1.030) | 0.148 | | |
| Lymphocyte (%) | 0.986 (0.967, 1.005) | 0.136 | | |
| Glucose (mg/dL) | 1.004 (1.002, 1.007) | 0.002 | 0.994 (0.987, 1.001) | 0.081 |
| BUN (mg/dL) | 1.064 (1.041, 1.086) | < .001 | 1.019 (0.962, 1.080) | 0.522 |
| Uric acid (mg/dL) | 1.170 (1.012, 1.352) | 0.034 | 0.701 (0.490, 1.004) | 0.052 |
| Total cholesterol (mg/dL) | 0.989 (0.981, 0.996) | 0.003 | 1.008 (0.989, 1.027) | 0.393 |
| Creatinine kinase (U/L) | 1.001 (0.997, 1.004) | 0.761 | | |
| CK-MB (ng/mL) | 1.001 (0.998, 1.005) | 0.459 | | |
| Troponin T (pg/mL) | 1.013 (0.998, 1.028) | 0.082 | | |
| NT pro BNP (pg/mL) | 1.010 (1.007, 1.014) | < .001 | 1.002 (0.992, 1.011) | 0.753 |
| Triglyceride (mg/dL) | 0.998 (0.994, 1.002) | 0.268 | | |
| HDL-Cholesterol (mg/dL) | 0.974 (0.946, 1.003) | 0.074 | | |
| LDL-Cholesterol (mg/dL) | 0.979 (0.970, 0.989) | < .001 | 0.976 (0.950, 1.002) | 0.069 |
| CRP (mg/L) | 1.046 (0.996, 1.100) | 0.074 | | |
| **Use of nephrotoxic medication** | | | | |
| Before[a] ACEI/ARB | 1.756(0.873, 3.533) | 0.115 | | |
| Before Beta blocker | 1.398(0.531, 3.679) | 0.498 | | |
| Before Statin | 1.443(0.763, 2.726) | 0.259 | | |

(*Continued*)

**Table 3.** (Continued)

| Variable | Univariable analysis | | Multivariable analysis | |
|---|---|---|---|---|
| | Odds Ratio (95% CI) | p-value | Odds Ratio (95% CI) | p-value |
| Before Insulin | 12.904(2.541, 65.530) | 0.002 | >999.999 (<0.001, >999.999) | 0.987 |
| Before oral DM | 2.441(1.342, 4.442) | 0.004 | 3.172 (0.327, 30.792) | 0.320 |
| Before NSAID | 0.705(0.034, 14.715) | 0.821 | | |
| After[b] ACEI /ARB | 0.150(0.079, 0.283) | < .001 | 0.738 (0.207, 2.625) | 0.638 |
| After Beta blocker | 0.887(0.438, 1.797) | 0.739 | | |
| After Statin | 0.412(0.165, 1.032) | 0.058 | | |
| After Insulin | 5.528(3.104, 9.847) | < .001 | 2.480 (0.561, 10.959) | 0.231 |
| After oral DM | 0.622(0.243, 1.598) | 0.324 | | |
| After NSAID | 0.500(0.026, 9.740) | 0.648 | | |

The effects of variables were analyzed using multivariable logistic regression.

PC-AKI, post-contrast acute kidney injury; CI, confidence interval; eGFR, estimated glomerular filtration rate; CAD, coronary artery disease; CKD, chronic kidney disease; LVEF, left ventricle ejection fraction; NA, not applicable; ref, reference; IABP, intra-aortic balloon pumping; ACEI, angiotensin-converting enzyme inhibitor; ARB, angiotensin II receptor blocker; DM, Diabetes mellitus; NSAID, nonsteroidal anti-inflammatory drug

Before[a]: Medication history before pPCI

After[b]: Medication history within 3 days after pPCI

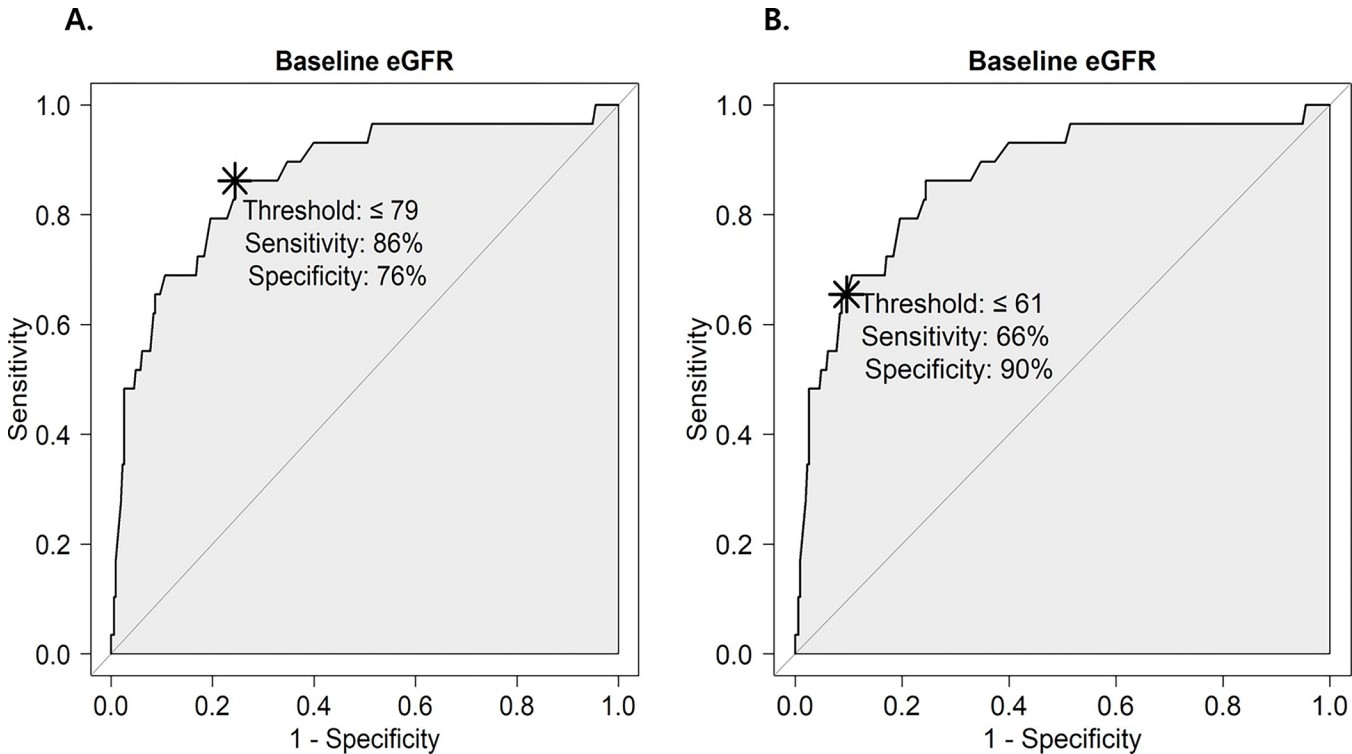

**Fig 2. A. Receiver operating characteristic (ROC) curve showing the probability of PC-AKI.** Cut-off point for eGFR $\leq$ 79mL/min/1.73 m$^2$. AUC (area under the curve) = 0.866 (0.789,0.925). **B. ROC curve was drawn for PC-AKI prediction probabilities using significant variables from the multivariable analysis.** Cut-off point for eGFR $\leq$ 61mL/min/1.73 m$^2$. AUC = 0.951 (0.924,0.979). PC-AKI, post-contrast acute kidney injury; eGFR, estimated glomerular filtration rate.

**Table 4. Comparison of previous and current diagnostic criteria according to baseline eGFR.**

| Baseline eGFR (mL/min/1.73 m$^2$) | Previous PC-AKI criteria | Current PC- AKI criteria | |
|---|---|---|---|
| | PC-AKI (n = 62) n (%) | PC-AKI (n = 59) n (%) | p-value |
| < 30 | 5 (8.1%) | 7 (11.9%) | 0.157 |
| 30–44 | 15 (24.2%) | 19 (32.2%) | 0.046 |
| 45–61 | 16 (25.8%) | 18 (30.5%) | 0.414 |
| 62–74 | 7 (11.3%) | 6 (10.2%) | 0.317 |
| 75–89 | 7 (11.3%) | 8 (8.5%) | 0.157 |
| ≥ 90 | 12 (19.4%) | 4 (6.8%) | 0.005 |

eGFR, estimated glomerular filtration rate, PC-AKI, post contrast- acute kidney injury.

We found that according to the previous and current diagnostic criteria for PC-AKI, 58.1% and 74.6% of the patients with an eGFR of ≤ 61 mL/min/1.73 m$^2$ had PC-AKI, respectively. The previous criteria led to a higher incidence of PC-AKI than that with the current criteria when the baseline eGFR was ≥ 90 mL/min/1.73 m$^2$ (p = 0.005). The current criteria led to a higher incidence of PC-AKI than that with the previous criteria when the baseline eGFR was between 30–44 mL/min/1.73 m$^2$ (p = 0.046) (Table 4).

In the mediation analysis with survival to discharge as the outcome, for both the previous and current diagnostic criteria, patients with an eGFR of ≤ 61 mL/min/1.73 m$^2$ had a lower probability of survival to discharge than those with an eGFR of > 61 mL/min/1.73 m$^2$ (p < .001, p < .001). The PC-AKI-mediated ratio of the total effect of the treatment on the rate of survival to discharge of patients with an eGFR of ≤ 61 was 73.63% when analyzed using the current diagnostic criteria and 53.92% when analyzed using the previous diagnostic criteria (Table 5).

When we analyzed the rate of survival to discharge based on the current PC-AKI criteria, the survival to discharge probability of patients with PC-AKI was lower than that of patients without PC-AKI, with an OR of 0.358 (0.173, 0.678, p = 0.003). After adjusting for major confounding variables, including eGFR, cardiogenic shock, and heart failure after exposure to the contrast medium, the survival probability reduced further at an OR of 0.253 (0.131, 0.480, p<0.001).

**Table 5. Comparison of previous and current diagnostic criteria by confirming the association of survival discharge for patients with eGFR ≤ 61 mediated through the occurrence of PC-AKI.**

| Mediation factor | Effect type | Odds Ratio (95% CI) | p-value | Prop. mediated |
|---|---|---|---|---|
| Previous PC-AKI | | | | |
| | Direct | 0.422 (0.19, 0.935) | 0.034 | |
| | Mediation | 0.365 (0.227, 0.579) | < .001 | 53.92% |
| | Total | 0.152 (0.074, 0.306) | < .001 | |
| Current PC-AKI | | | | |
| | Direct | 0.611 (0.243, 1.532) | 0.294 | |
| | Mediation | 0.253 (0.131, 0.48) | < .001 | 73.63% |
| | Total | 0.152 (0.074, 0.306) | < .001 | |

PC-AKI, post contrast acute kidney injury; Prop. Mediated, PC-AKI-mediated ratio of total effects on survival to discharge in patients with eGFR ≤ 61.

## Discussion

The primary objective of this study was to identify the risk factors that increase the incidence of PC-AKI in patients who undergo PCI for STEMI in the ED and to determine the reference value of baseline renal function for the same. We found that the probability of PC-AKI occurrence increased when the baseline eGFR was $\leq 79$ mL/min/1.73 m$^2$ in our patient group. Furthermore, the optimal cutoff value of baseline eGFR that indicated the risk of PC-AKI after adjusting for statistically significant risk factors was eGFR of $\leq 61$ mL/min/1.73 m$^2$. The optimal cutoff value of baseline eGFR can be used as a reference for healthcare professionals to prevent or manage PC-AKI in patients with STEMI who require PCI. Our results also demonstrated that the occurrence of PC-AKI after PCI for STEMI was associated with a decreased survival rate.

It is well known that pre-existent CKD is the most important pre-procedure risk factor for PC-AKI. In 1999, the European Society of Urogenital Radiology Consensus Working Panel reported that PC-AKI represents a clinically significant risk when the baseline concentration of SCr is $\geq 1.0$ mg/dL ($\geq 88.4$ mmol/L) in women and $\geq 1.3$ mg/dL ($\geq 115$ mmol/L) in men [22]. These values approximate to an eGFR of $< 60$ mL/min/1.73 m$^2$, as in the definition for CKD stages 3–5. Until now, this value has been generally recognized as the threshold for the risk of PC-AKI [23]. Cardiogenic shock is defined as a systolic blood pressure (SBP) <90 mmHg for $\geq 30$ min or use of pharmacological and/or mechanical support to maintain an SBP $\geq 90$ mmHg paired with evidence of end-organ hypoperfusion, which is also a risk factor for PC-AKI. The contrast medium-induced renal ischemia from renal hypoperfusion is caused by cardiogenic shock (AMI), excessive vasodilation (anaphylaxis), or intravascular volume depletion (dehydration, hemorrhage or sepsis) [24]. Impaired cardiac function, such as recent history of pulmonary edema, or LVEF $< 45\%$, AMI, or the presence of congestive heart failure classified according to New York Heart Failure Association III or IV, has been known to be an independent risk factor of the development of PC-AKI [25,26]. As ACEIs and ARBs have their well-established benefit, they have been widely used to increase renal protection in patients with diabetes mellitus and to reduce morbidity and mortality in patients with coronary artery disease. However, AKI can be induced by ACEIs in patients with impaired glomerular filtration, depending on the angiotensin II-mediated efferent vascular tone, for instance in patients with renal artery disease, heart failure, or severe volume depletion [27]. Although the inhibition of ACE can contribute to the occurrence of AKI in specific conditions, it remains controversial whether these medications should be discontinued before coronary angiography to minimize PC-AKI. The Kidney Disease-Improving Global Outcomes guidelines proposed that there is insufficient evidence to recommend discontinuation of these medications for patients receiving injections of contrast medium [28].

We did not find any statistically significant association between the use of ACEi/ARB after PCI and the development of PC-AKI in our study. The risk factors for developing PC-AKI in this study were similar to those reported in previous studies.

Although there was no significant difference in the amount of contrast medium between the group with PC-AKI and the group without PC-AKI in this study, Shrivastava et al. demonstrated that the ultra-low contrast PCI protocol was reasonably safe and effective in reducing the incidence of CI-AKI in a high-risk cohort compared to conventional PCI [29]. Despite the significance of the type and amount of contrast medium in the development of PC-AKI, our results suggest that the etiology of AKI in STEMI patients undergoing PCI is multifactorial, including the role of contrast medium, rather than solely attributable to contrast medium alone. This finding aligns with the results of a previous study [30].

In addition, incorporating both pre- and post-procedural Mehran 2 risk scores could be valuable in determining patient risk prior to percutaneous coronary intervention (PCI).

However, in our study, all patients were categorized into a single group comprising only STEMI patients. Unfortunately, we were unable to collect the required variables, such as procedural bleeding and no flow, to calculate the risk score after the procedure. The large prospective study is needed to validate pre- and post-procedural Mehran 2 risk scores in patients undergoing PCI for STEMI [31].

The 2018 ESUR update identified an eGFR of $<30$ ml/min/1.73 m$^2$ as an important risk factor for PC-AKI before the administration of intravenous or intra-arterial contrast medium with second-pass renal exposure. Myung et al. reported a significant increase in the incidence of PC-AKI in patients with ischemic stroke who underwent CTA and cerebral angiography with a baseline GFR of $< 43$ mL/min/1.73 m$^2$ [32]. Several studies have shown that the risk of PC-AKI is high with eGFR $\leq 45$ mL/min/1.73 m$^2$ and intravenous administration of contrast medium. However, an eGFR cutoff of $< 60$ mL/min/1.73 m$^2$ is preferred for intra-arterial administration of contrast medium [23].

The present study proposed slightly higher baseline eGFR cutoff values of $\leq 61$ mL/min/1.73 m$^2$ with multivariable adjustment, respectively, than those in other studies without multivariable adjustment. Specially, the 2018 ESUR guidelines propose that the risk of contrast-induced acute kidney injury (PC-AKI) is higher when the eGFR is $<45$ mL/min/1.73 m$^2$, without considering various risk factors after simple coronary angiography [20]. Patients with STEMI receiving PCI had many risk factors for PC-AKI and can be attributed to the large amount of contrast medium they received through intra-arterial injection. Therefore, physicians should exercise caution regarding the high risk of PC-AKI before and after using contrast agents in patients who undergo PCI for STEMI in the ED. Until now, it has not been possible to determine an optimal cutoff for the development of PC-AKI, specifically in patients with STEMI, which is a more severe condition compared to other diseases. This study, being the first to suggest an optimal eGFR cutoff for predicting the occurrence of PC-AKI in patients with STEMI while considering various risk factors. The clinical implications of our results are significant, as they could inform clinical decisions on the use of contrast-enhanced imaging procedures for patients with reduced kidney function or several high-risk factors for PC-AKI.

The current diagnostic criteria for PC-AKI use a lower eGFR value than the previous criteria. To avoid overestimating the incidence of AKI using the new definition of PC-AKI, the criteria were based on a lower baseline eGFR for predicting the occurrence of PC-AKI; this was accomplished by decreasing the absolute reference value of creatinine and increasing the relative reference value of creatinine. This study aimed to provide clinical evidence for the newly revised criteria of PC-AKI by comparing the incidence rates of PC-AKI among patients who underwent PCI for STEMI based on both the previous and current diagnostic criteria. As per the previous diagnostic criteria, 58.1% of patients with an eGFR $\leq 61$ mL/min/1.73 m$^2$ developed PC-AKI, and the incidence of PC-AKI was higher when the baseline eGFR was $\geq 90$ mL/min/1.73 m$^2$. As per the current diagnostic criteria, 74.6% of patients with an eGFR $\leq 61$ mL/min/1.73 m$^2$ developed PC-AKI, and the occurrence of PC-AKI was higher when the baseline eGFR was 30–44 mL/min/1.73 m$^2$. These results suggest that the current diagnostic criteria more accurately reflect the patient group with impaired renal function.

A mediation analysis was performed with survival to discharge as the outcome for both the previous and current diagnostic criteria. The patient group with an eGFR $\leq 61$ mL/min/1.73 m$^2$ had a lower probability of survival to discharge than the patient group with an eGFR $>61$ mL/min/1.73 m$^2$ (p $<$ .001 for both). When PC-AKI was selected as the mediator in our mediation analysis, the results based on the current diagnostic criteria showed a lower OR for survival discharge than the results based on the previous diagnostic criteria (previous criteria, OR, 0.365 (95% CI, 0.227–0.579 vs. current criteria, OR, 0.253 (0.131–0.480; p $<$ .001 for both). As per the mediation analysis based on the current diagnostic criteria, the PC-AKI-mediated ratio

of the total effect the treatment on the rate of survival to discharge of patients with an eGFR $\leq$ 61 was 73.63%, which was higher than the 53.92% obtained with the previous diagnostic criteria. These findings suggest that the current diagnostic criteria offer more reliable diagnostic and prognostic data in terms of the clinical outcome of patients who undergo PCI for STEMI as well as allow a more accurate prediction of the occurrence of PC-AKI than the previous diagnostic criteria.

Finally, to date, various preventive methods for PC-AKI, other than fluid therapy, are being studied. Adequate hydration is the primary approach for preventing PC-AKI; however, additional measures such as high-dose statins (as indicated for secondary prevention), optimal hemodynamics, and discontinuation of nephrotoxic drugs, might offer significant benefits. In the future, large-scale prospective studies are needed to evaluate the effectiveness of these additional preventive measures beyond fluid therapy.

## Limitations

This study has several noteworthy limitations. First, given the retrospective nature of the research design, the effect of confounders that were not identified or controlled for as well as selection bias cannot be denied. Second, the results may have limited generalizability since the study was conducted at a single center. Third, we excluded patients who could be considered severely ill (such as those who died within 48 hours after examination, those who were discharged or transferred within 48 hours, and those without creatinine results within 72 hours) in order to evaluate a condition in which renal function impairment occurs within 3 days following the intravascular administration of a contrast medium, based on the definition of PC-AKI. However, this exclusion criterion may have introduced additional bias. Finally, this study focused on collecting data on serum creatinine (Cr) and estimated glomerular filtration rate (eGFR) within a 3-day period following percutaneous coronary intervention (PCI), with contrast serving as the diagnostic criterion for PC-AKI. PCI has the potential to enhance both cardiac and renal hemodynamics, which may contribute to the recovery of renal function. Additionally, PCI can play a beneficial role in preventing the occurrence of PC-AKI. However, to further validate the impact of PCI on renal function improvement through hemodynamic recovery and the prevention of PC-AKI development, it is essential to conduct large-scale prospective studies over a longer duration.

## Conclusions

This study compared the incidence of PC-AKI between the previous and current diagnostic criteria among patients who underwent PCI for STEMI. The current diagnostic criteria more accurately identified patients with impaired renal function and enhanced the reliability of the diagnosis and prognosis. Patients with a baseline eGFR of $\leq$ 79 mL/min/1.73 m$^2$ had a higher incidence of PC-AKI, and after adjusting for risk factors, the risk of PC-AKI was increased in patients with a baseline eGFR of $\leq$ 61 mL/min/1.73 m$^2$. These findings can help physicians identify patients at a high risk of developing PC-AKI and to exercise caution before and after using intra-arterial contrast agents. Third, there may be additional bias due to the exclusion of seriously ill patients who died within 48 hours after examination, patients who were discharged or transferred within 48 hours, and patients without creatinine results within 72 hours after examination.

## Supporting information

**S1 File. Survival discharge according to PC-AKI occurrence.**
(DOCX)

**S2 File. Anonymized data set for the study.**
(XLSX)

## Author Contributions

**Conceptualization:** Je Sung You, Jin Ho Beom.

**Data curation:** Jin Ho Beom.

**Investigation:** Junho Cho.

**Methodology:** Je Sung You, Jin Ho Beom.

**Software:** Je Sung You, Hye Jung Shin, Jin Ho Beom.

**Supervision:** Je Sung You, Jin Ho Beom.

**Validation:** Hye Jung Shin.

**Visualization:** Junho Cho.

**Writing – original draft:** Jin Ho Beom.

**Writing – review & editing:** Je Sung You, Jin Ho Beom.

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
