## [Decision Letter · Decision Letter 0]

9 May 2023

PONE-D-23-10724Baseline eGFR cutoff for increased risk of post-contrast acute kidney injury in patients undergoing percutaneous coronary intervention for ST-elevation myocardial infarction in the emergency departmentPLOS ONE

Dear Dr. Beom,

Thank you for submitting your manuscript to PLOS ONE. After careful consideration, we feel that it has merit but does not fully meet PLOS ONE’s publication criteria as it currently stands. Therefore, we invite you to submit a revised version of the manuscript that addresses the points raised during the review process.

We look forward to receiving your revised manuscript.

Kind regards,

Satoshi Higuchi

Academic Editor

PLOS ONE

Journal Requirements:

Reviewers' comments:

Reviewer's Responses to Questions

**Comments to the Author**

1. Is the manuscript technically sound, and do the data support the conclusions?

Reviewer #1: Yes

Reviewer #2: Yes

2. Has the statistical analysis been performed appropriately and rigorously? 

Reviewer #1: Yes

Reviewer #2: Yes

3. Have the authors made all data underlying the findings in their manuscript fully available?

Reviewer #1: Yes

Reviewer #2: Yes

4. Is the manuscript presented in an intelligible fashion and written in standard English?

Reviewer #1: Yes

Reviewer #2: Yes

5. Review Comments to the Author

Reviewer #1: The authors conducted an interesting study focused on STEMI patients at risk of contrast-induced acute kidney injury (CI-AKI) in the setting of the emergency department.

Using a contemporary retrospective cohort analysis of 754 patients with STEMI undergoing primary PCI and enrolled in the Fast Interrogation Rule for STEMI critical pathway program they determined the optimal cutoff for baseline eGFR to identify patients at high risk for CI-AKI, namely 73 mL/min/1.73m2 after multivariable adjustment.

Although it’s well known that PCI for acute coronary syndromes is a major risk setting for CI-AKI, most of the previous studies focused on the general population undergoing PCI and the added value for this study is the inclusion of STEMI-only patients.

Yet, there are a few concerns about the paper to be addressed:

1. The term “post-contrast acute kidney injured (PC-AKI)” should be changed with the most used and well-known “contrast-induced acute kidney injury (CI-AKI)” across the paper.

2. In the introduction is stated that there are no effective therapeutic modalities for the prevention of PC-AKI but intravenous volume expansion with normosaline is the recommended prevention strategy according to US and European guidelines, even if the evidence from randomized clinical trials and meta-analysis are conflicting. A tailored hydration protocol according to LVEDP has proven to be effective in the POSEIDON trial and those findings were confirmed in a recent meta-analysis from Michel P et al (Meta-analysis of intravascular volume expansion strategies to prevent contrast-associated acute kidney injury following invasive angiography. Catheter Cardiovasc Interv. (2021) 98:1120–32. doi: 10.1002/ccd.29387). Moreover, the protective effect of statins has been consistently demonstrated in multiple clinical trials and meta-analyses, therefore is recommended to soften the sentence.

3. The authors should provide explanations on the very long average time-to-PCI that far exceeds the recommended 60 minutes for a primary PCI center. This is potentially relevant for the occurrence of CI-AKI as delayed reperfusion increases the incidence of post-PCI slow-flow or no-flow and the kidney exposure time to hypoperfusion due to low cardiac output.

4. In Table 1 White blood cells should be abbreviated without punctuation.

5. In the logistic regression analysis, the baseline eGFR and the age have been categorized, likely to improve the clinical understanding of the results. However, categorizing a continuous variable is generally non recommended as important information could be neglected. Have the authors tried to use continuous variables as predictors? Considering the very large confidence intervals this could have improved the model. Moreover, is not useful to add the variable “pre-existing CKD” in the multivariable model already having the eGFR categories in the CKD range. It’s usually better to use a more parsimonious model without variables with overlapping information but the authors included 13 variables with just 5 statistically significant.

6. Since the introduction of the Mehran Score in 2004, eGFR has been recognized as a major predictor of CI-AKI. More recently in 2021 prof. Mehran updated her historical score with a contemporary, simple, and validated version using a large cohort of 14616 patients. In the model, many important clinical variables resulted significant predictors of CI-AKI like STEMI at presentation, EF<40%, hemoglobin <11 g/dL, basal glucose >150 mg/dL, and even procedural details like contrast volume, procedural bleeding, slow-flow or no-flow and complex anatomies. In this study, some of the important variables like hemoglobin, glucose, and contrast amount are not significantly different in the two groups. Have these variables been investigated in the multivariable model given their established relationships with CI-AKI? In the discussion, there should be a comparison between the results of this study and the data from Prof. Mehran.

7. The highest impact of CI-AKI on survival is noticeable at 30 days. Are data at 30 days available at least for a group of patients?

8. In the discussion there should be more attention to the clinical significance of having a higher cut-off of 73 mL/min/1.73m2 that independently predicts the onset of CI-AKI for STEMI patients. What are the implications of classifying a wider population at risk of contrast nephropathy than the conventional threshold of 60 mL/min/1.73m2? How this new cut-off is actionable and what strategies should be implemented in the ED once patients at risk have been identified?

9. Line 378, second sentence: correct the typo “Firs” with “First”.

Reviewer #2: ACS is a high risk state due to highly thrombogenic state, increased inflammation, and decreased renal perfusion through vasoconstriction or hemodynamic instability. Where exposure of contrast medium can increase chances of CI AKI.

In the present paper, the authors present their data of optimal cut-offs of GFR for predicting CI-AKI. 

The introduction can be better rephrased. Following are some point regarding the same.

Comment 1: line 51: pPCI is a high-risk procedure for post-contrast acute kidney injury (PC-AKI). Here we can modify as - ACS is a high risk state due to highly thrombogenic state, increased inflammation, and decreased renal perfusion through vasoconstriction or hemodynamic instability. Where exposure of contrast medium can increase chances of CI AKI. Authors can give quotations of studies reporting such increased incidences of AKI in pPCI.

Comment 2: line 56: Unlike CT- kindly use full form

Comment 3: for PCI, the medium: use contrast

Comment 4: ‘Although some reports suggest that hydration can reduce PC-AKI in patients with STEMI, to our knowledge, there are no effective therapeutic modalities for the prevention of PC-AKI.’ – Pre and post PCI hydration is the only intervention which has shown to reduce incidences of CI-AKI. Kindly give citiations too.

Comment 5: ‘Studies have shown that certain patient groups with baseline renal function, such as those with estimated glomerular filtration rates (eGFRs) <30, <45, or <60, are at an increased risk of developing PC-AKI’ – Kindly use 1 cutoff and their citation. Mostly <60 has been universally accepted as upper cut-off. 

Comment 6: line 304: Hypotension : cardiogenic shock

Comment 7: Line 324 ‘The fact that there was no significant difference in the contrast mediums amount between the group with PC-AKI and the group without PC-AKI in this study also supports these results’ – This conclusion is oversimplification of results. Studies have proved that the lower the contrast we use the less are the chances of developing CI-AKI and thus the advent of ultra-low contrast PCI in such high risk low GFR cases. There have been recently more robust data published which further strengthen the case for a ULC-PCI. PMID: 36007555. These data can be added to further enhance the evidence base for the same and improve the impact of the article. 

Comment 8: Addition of pre-procedural risk scores and post-procedure risk scores can be done which help objectively in decision making regarding the risk of the patient before PCI. The patietts could have been classified according to their risk scores and would have given a more deeper insight on the causes of development od AKI in an individual.

Comment 9: The average contrast consumption in the setting of pPCI was more than 100 ml which is acceptable in a pPCI setting. However what do the authors conclude in a stting of a low GFR what should the interventionalist do. Should he give less contrast or give more hydration? This articles conclusion seems there is no association with the amount of contrast use so such an exercise would prove futile. Kindly elaborate in discussion as to what this article and these cut-offs mean in real world.

Comment 10: Also what was the follow-up GFR of patients post intervention. Does primary PCI help the GFR in the longer run. A longer follow-up would have helped. As reperfusion helps improve cardiac hemodynamics and renal hemodynamics in the patients. So is pPCI benificial to the patient renal wise . what are authors opinions regarding it. 

Comment 11: what are authors views after the study as to which patients should receive ACEi and ARNis and on what day after intervention based on their data. Any expert comments as to minimize the risk of AKI and also utilize them for their benifits

From Editor

Thank you for the opportunity to consider your work.

The current study was evaluated retrospectively, but the authors stated that data acquisition was conducted prospectively. In general, written informed consent should not be waived in prospective registry.

Some patients in the setting of STEMI present with AKI due to unstable hemodynamics. How did the authors distinguish such patients from those who developed PC-AKI?

The mediation analysis conducted in the current study seems to be inappropriate because eGFR and PC-AKI can demonstrate an interaction. Killip IV or cardiogenic shock may be a potential exposure variable and PC-AKI can be an mediator. It would be difficult to demonstrate a causal-relationship between PC-AKI and in-hospital mortality using the current registry.

Please clarify usage rates of ECMO and Impella as well as a prevalence of cardiac arrest.

Some clinical values such as the average time between admission and pPCI, CK/CK-MB, NT-proBNP, and so on were described inappropriately. They do not follow a normal distribution. Please reconfirm demographic and exam data carefully.

In Table1, does “before ACEI/ARB” mean ACEI/ARB before pPCI? Please make such description easy to understand.

6. PLOS authors have the option to publish the peer review history of their article (what does this mean?). If published, this will include your full peer review and any attached files.

Reviewer #1: No

Reviewer #2: **Yes: **Abhinav Shrivastava

---

## [Author Response · Author response to Decision Letter 0]

2 Aug 2023

Response to Editor comments

Thank you for the opportunity to consider your work.

Response: We appreciate your kind comments and useful suggestions regarding our manuscript.

The current study was evaluated retrospectively, but the authors stated that data acquisition was conducted prospectively. In general, written informed consent should not be waived in prospective registry.

Response: Thank you for your comment. We apologize for the confusion. As stated in the materials and methods section of the manuscript, this was a retrospective observational cohort study. However, we used data from a prospective registry to demonstrate that, when emergent patients with STEMI visited the emergency department, the activation of critical pathways was prospectively and sequentially performed based on a predetermined protocol and that the relevant data were accumulated in electronic health records (EMRs). Given that we retrospectively analyzed the data from this registry, obtaining informed consent was not required.

Some patients in the setting of STEMI present with AKI due to unstable hemodynamics. How did the authors distinguish such patients from those who developed PC-AKI?

Response: Thank you kindly for your comment. According to the guidelines, the concept of PC-AKI differs from that of CI-AKI. If AKI is solely caused by the use of contrast media, then the term CI-AKI is appropriate. However, in this study, we used the term PC-AKI to demonstrate a significant deviation from the suggested renal function cutoff value after contrast agents are administrated to patients with risk factors that increase the likelihood of developing AKI.

We referred to unstable hemodynamics as cardiogenic shock. However, we did not conduct a separate analysis to determine whether PC-AKI was solely caused by the contrast agent or cardiogenic shock, as this differentiation was beyond the scope of our study. We found no significant difference in contrast volume between the group that developed PC-AKI and the group that did not. Therefore, in patients administered contrast media, it is believed that PC-AKI is influenced by multiple factors rather than the contrast media alone. Given that the likelihood of PC-AKI occurrence is higher in patients with unstable hemodynamics after contrast agents are employed, preventive measures should be carefully considered even if the renal function cutoff value is high. This study revealed that specific medications and hemodynamic instability can increase the risk of PC-AKI. In the future, we plan to conduct a large prospective study that comprehensively examines risk factors beyond contrast agents, and their impact on the occurrence of PC-AKI.

The mediation analysis conducted in the current study seems to be inappropriate because eGFR and PC-AKI can demonstrate an interaction. Killip IV or cardiogenic shock may be a potential exposure variable and PC-AKI can be an mediator. It would be difficult to demonstrate a causal-relationship between PC-AKI and in-hospital mortality using the current registry.

Response: We appreciate your comment. Accordingly, we discussed this with our statisticians and obtained the following results concerning the interaction between eGFR and PC-AKI.

[Confirmation of the interaction effect of AKI and baseline eGFR on survival discharge]

 OR (95% CI) Interaction p-value

Effect of baseline eGFR in the No PC-AKI group 1.037 (1.017, 1.058) 0.003

Effect of baseline eGFR in the PC-AKI group 0.989 (0.964, 1.014) 

eGFR, estimated glomerular filtration rate; PC-AKI, post-contrast acute kidney injury

The analysis results indicate that in the No PC-AKI group, for each 1-unit increase in baseline eGFR, there was a significant 1.037-fold increase in the odds of survival discharge, as evidenced by the confidence interval not including 1. However, in the “PC-AKI” group, the 95% confidence interval for the effect of baseline eGFR on survival discharge includes 1, indicating no significance. With an interaction p-value of 0.003, it can be concluded that there is a significant difference in the baseline eGFR effect on survival discharge based on the presence of AKI. Therefore, the mediation analysis of eGFR and PC-AKI conducted in this study appears to be appropriate.

In addition, according to the Editor’s suggestion, we conducted an analysis considering Killip IV or cardiogenic shock as the exposure variable and PC-AKI as the mediator, to examine the impact of PC-AKI on survival discharge. The results revealed that, when PC-AKI occurred, Killip classes III and IV were associated with a decrease in survival discharge by 0.318 and 0.388 times, respectively. Particularly, Killip class III showed a significant association with survival discharge only when PC-AKI occurred, indicating that PC-AKI plays an important role as a mediator variable.

In the pre-procedural status analysis, we observed that in cases of IABP, cardiogenic shock, and heart failure, PC-AKI occurrence was associated with a decreased odds ratio for survival discharge, with values of 0.301, 0.462, and 0.361, respectively. Particularly for cardiogenic shock and heart failure, the association with survival discharge was significant only when PC-AKI occurred, indicating that PC-AKI plays an important role as a mediator variable in these two factors.

In conclusion, PC-AKI exhibits a potential interaction effect with eGFR, and there are statistically significant differences, confirming the potential of conducting mediation analysis. Furthermore, when considering other variables as independent variables and PC-AKI as the mediator, the results of the mediation analysis revealed that PC-AKI can have a significant impact on survival discharge.

　 　 　 Indirect effect Direct effect Total effect Prop. mediated

Dependent variable Independent variable Mediator variable OR (95% CI) p-value OR (95% CI) p-value OR (95% CI) p-value 

Survival discharge Killip class (ref=1) PC-AKI ref - ref - ref - 

 Killip class II 0.727 (0.451, 1.114) 0.168 0.384 (0.049, 2.744) 0.351 0.168 (0.054, 0.519) 0.002 24.98%

 Killip class III 0.318 (0.176, 0.555) <.001 0.550 (0.019, 11.825) 0.716 0.139 (0.045, 0.432) 0.001 65.76%

 Killip class IV 0.388 (0.228, 0.632) <.001 0.234 (0.088, 0.715) 0.007 0.068 (0.028, 0.161) <.001 39.46%

Survival discharge Pre-procedural status (ref=NA) PC-AKI ref - ref - ref - 

 IABP 0.301 (0.138, 0.589) 0.001 0.207 (0.046, 0.984) 0.044 0.064 (0.020, 0.205) <.001 43.28%

 Cardiogenic shock 0.462 (0.285, 0.727) 0.001 0.341 (0.122, 1.092) 0.054 0.113 (0.049, 0.259) <.001 41.84%

 Heart failure 0.361 (0.190, 0.648) 0.001 0.716 (0.001, 243.629) 0.920 0.194 (0.051, 0.740) 0.016 75.30%

IABP, intra-aortic balloon pump, PC-AKI, post-contrast acute kidney injury

Please clarify usage rates of ECMO and Impella as well as a prevalence of cardiac arrest.

Response: Thank you for your suggestion. Unfortunately, Impella is not currently used in Korea due to the lack of permission for its use. In our study, a total of 40 patients were treated with ECMO. Among them, 17 patients (16 Out-of-hospital cardiac arrest and 1 In-hospital cardiac arrest) were excluded from the study based on a predetermined protocol (cardiac arrest). Among the 23 enrolled patients, 15.3% belonged to the PC-AKI group (9/59) while 2.0% belonged to the No PC-AKI group (14/695). We have added this information to Table 1 of the revised manuscript.

Some clinical values such as the average time between admission and pPCI, CK/CK-MB, NT-proBNP, and so on were described inappropriately. They do not follow a normal distribution. Please reconfirm demographic and exam data carefully.

Response: We appreciate your kind comment. At the beginning of the study, we consulted our statisticians regarding the statistical analysis. According to the central limit theorem, if the sample size (n) is large enough (typically greater than 30), the average will approximate a normal distribution. Accordingly, we performed the analysis using parametric methods. Our statisticians conducted the analysis in the same manner. We value your opinion on this matter. In accordance with your comment, we conducted normality tests and revised the demographic and test results, as well as the data in Table 1. Based on the revised results, we reconducted the univariable and multivariable analyses. While our overall results remain unchanged, there have been some updates. The optimal eGFR cutoff value has changed to 61 mL/min/1.73 m2. In particular, the cardiac function of patients with STEMI may be compromised, leading to limited fluid therapy. Considering various risk factors in these patients, determining the optimal cutoff for PC-AKI occurrence can be helpful in guiding fluid therapy as a preventive measure. Therefore, we believe that this information can be utilized as a guideline for fluid treatment to prevent AKI.

In Table1, does “before ACEI/ARB” mean ACEI/ARB before pPCI? Please make such description easy to understand.

Response: Thank you for your suggestion. We apologize for the unclear wording. To enhance clarity, we have revised the wording as follows: 

Before: Medication history before pPCI

After: Medication history within 3 days after pPCI

Table 1 and 3 has been amended accordingly.

Response to Reviewer 1 comments

Reviewer #1: The authors conducted an interesting study focused on STEMI patients at risk of contrast-induced acute kidney injury (CI-AKI) in the setting of the emergency department.

Using a contemporary retrospective cohort analysis of 754 patients with STEMI undergoing primary PCI and enrolled in the Fast Interrogation Rule for STEMI critical pathway program they determined the optimal cutoff for baseline eGFR to identify patients at high risk for CI-AKI, namely 73 mL/min/1.73m2 after multivariable adjustment.

Although it’s well known that PCI for acute coronary syndromes is a major risk setting for CI-AKI, most of the previous studies focused on the general population undergoing PCI and the added value for this study is the inclusion of STEMI-only patients.

Response: We appreciate your kind comments and useful suggestions regarding our manuscript.

Yet, there are a few concerns about the paper to be addressed:

1. The term “post-contrast acute kidney injured (PC-AKI)” should be changed with the most used and well-known “contrast-induced acute kidney injury (CI-AKI)” across the paper.

Response: We agree with your opinion; however, each set of guidelines slightly differs when it comes to using the term. In this study, we referred to the 2018 recommendations for updated ESUR Contrast Medium Safety Committee guidelines that mention the following definition:

“The preferred term for acute kidney injury associated with CM administration when no control population is available is Post-Contrast Acute Kidney Injury (PC-AKI). The term Contrast-Induced Acute Kidney Injury (CI-AKI) should be used only when comparison with a control allows CM to be shown to be the cause of the acute kidney injury.”

In this study, we found that the use of contrast media increased the risk of AKI when multiple risk factors for AKI were present. The study does not propose that the sole cause of AKI is the contrast media; therefore, the term PC-AKI is used, which is more appropriate given the context.

Reference:

Van der Molen AJ, Reimer P, Dekkers IA, Bongartz G, Bellin MF, Bertolotto M, et al. Post-contrast acute kidney injury - Part 1: definition, clinical features, incidence, role of contrast medium and risk factors: recommendations for updated ESUR Contrast Medium Safety Committee guidelines. Eur Radiol. 2018;28: 2845–2855.

2. In the introduction is stated that there are no effective therapeutic modalities for the prevention of PC-AKI but intravenous volume expansion with normosaline is the recommended prevention strategy according to US and European guidelines, even if the evidence from randomized clinical trials and meta-analysis are conflicting. A tailored hydration protocol according to LVEDP has proven to be effective in the POSEIDON trial and those findings were confirmed in a recent meta-analysis from Michel P et al (Meta-analysis of intravascular volume expansion strategies to prevent contrast-associated acute kidney injury following invasive angiography. Catheter Cardiovasc Interv. (2021) 98:1120–32. doi: 10.1002/ccd.29387). Moreover, the protective effect of statins has been consistently demonstrated in multiple clinical trials and meta-analyses, therefore is recommended to soften the sentence.

Response: Thank you kindly for your comment. We agree with your opinion and have revised the Introduction according to your recommendations. 

“The incidence of PC-AKI is high in patients undergoing cardiac procedures such as PCI [6,13]. Although some reports suggest that hydration can reduce PC-AKI in patients with STEMI, to our knowledge, there are no effective therapeutic modalities for the prevention of PC-AKI. Intravascular volume expansion can effectively prevent PC-AKI following invasive angiography. Numerous studies have been conducted to prevent PC-AKI, employing various strategies, such as appropriate risk stratification, reducing iodinated contrast dose, utilizing low or iso-osmolar contrast media, withholding nephrotoxic medications when necessary, and administering protective agents [14,15]. However, the use of contrast media is associated with multiple adverse effects; hence, it is a burden on physicians in the clinical setting [7].”

References:

14.Michel P, Amione-Guerra J, Sheikh O, Jameson LC, Bansal S, Prasad A. Meta-analysis of intravascular volume expansion strategies to prevent contrast-associated acute kidney injury following invasive angiography. Catheter Cardiovasc Interv. 2021;98: 1120–1132.

15.Gupta RK, Bang TJ. Prevention of contrast-induced nephropathy (CIN) in interventional radiology practice. Semin Intervent Radiol. 2010;27: 348–359.

3. The authors should provide explanations on the very long average time-to-PCI that far exceeds the recommended 60 minutes for a primary PCI center. This is potentially relevant for the occurrence of CI-AKI as delayed reperfusion increases the incidence of post-PCI slow-flow or no-flow and the kidney exposure time to hypoperfusion due to low cardiac output.

Response: We appreciate your opinion and perspective. We agree that the probability of PC-AKI occurrence can increase if the door-to-balloon time is prolonged. However, upon analyzing the relevant data, we believe that the mean door-to-balloon time became longer due to a few cases in both groups where PCI was performed very late. The median door-to-balloon time in the No PC-AKI group was 66 minutes [IQR 53, 86], while it was 85 minutes [IQR 65, 135] in the PC-AKI group—this is close to the door-to-balloon time recommended in the treatment guidelines. The median door-to-balloon time in all patients was 68 minutes [IQR 54, 89]. In both groups, there were cases where PCI was delayed based on the judgment of a cardiologist for patients with delayed implementation of PC-AKI. Initially, there were no STEMI findings on the ECG; however, as the ECG started to reveal STEMI findings while the patients were in the emergency department, the delayed critical pathway was activated. However, in this study, the p-value for door-to-balloon time was 0.451 in the No PC-AKI and PC-AKI groups, indicating no significant difference between the groups. Therefore, it is difficult to conclude that door-to-balloon time significantly affected the occurrence of PC-AKI. Based on your comment, we have revised door-to-balloon time from the mean to the median value in Table 1.

4. In Table 1 White blood cells should be abbreviated without punctuation.

Response: Thank you for your suggestion. We have revised accordingly by changing “W.B.C.” to “WBC”.

5. In the logistic regression analysis, the baseline eGFR and the age have been categorized, likely to improve the clinical understanding of the results. However, categorizing a continuous variable is generally non recommended as important information could be neglected. Have the authors tried to use continuous variables as predictors? Considering the very large confidence intervals this could have improved the model. 

Response: We have discussed your valuable suggestion with our statisticians and consequently reanalyzed age and eGFR as continuous variables. Tables 1 and 3 have been amended accordingly.

Moreover, is not useful to add the variable “pre-existing CKD” in the multivariable model already having the eGFR categories in the CKD range. It’s usually better to use a more parsimonious model without variables with overlapping information but the authors included 13 variables with just 5 statistically significant.

Response: Thank you kindly for your comment, which we have discussed with our statisticians. We acknowledge the possibility of statistical over- or underestimation due to the increase in variables. However, we believe that a previous history of chronic renal failure, which provides information about pre-existing renal function, and the eGFR classification, which reflects renal function at the time of the study, represent variables of slightly different natures. For instance, during the data collection period, we observed that patients with pre-existing CKD had an eGFR >60 mL/min/1.73 m², while patients without pre-existing CKD exhibited an eGFR <60 mL/min/1.73 m². In other words, these two variables correspond to different time points. Therefore, it is important to consider the CKD and eGFR category variables together in statistical analyses.

6. Since the introduction of the Mehran Score in 2004, eGFR has been recognized as a major predictor of CI-AKI. More recently in 2021 prof. Mehran updated her historical score with a contemporary, simple, and validated version using a large cohort of 14616 patients. In the model, many important clinical variables resulted significant predictors of CI-AKI like STEMI at presentation, EF<40%, hemoglobin <11 g/dL, basal glucose >150 mg/dL, and even procedural details like contrast volume, procedural bleeding, slow-flow or no-flow and complex anatomies. In this study, some of the important variables like hemoglobin, glucose, and contrast amount are not significantly different in the two groups. Have these variables been investigated in the multivariable model given their established relationships with CI-AKI? In the discussion, there should be a comparison between the results of this study and the data from Prof. Mehran. 

Response: Thank you for your valuable comment. We agree with your opinion about the many important clinical variables in PC-AKI. At the beginning of the study, we consulted our statisticians regarding the statistical analysis. According to the central limit theorem, if the sample size (n) is large enough (typically greater than 30), the average will approximate a normal distribution. Accordingly, we performed the analysis using parametric methods. Our statisticians conducted the analysis in the same manner. However, in response to the Editor’s request, we conducted normality tests and revised the demographic and test results accordingly.

We highly value your perspective on this matter. Based on your comment, we held discussions with our statisticians and proceeded to conduct a revised statistical analysis, incorporating the significant clinical variables you mentioned. As a result, we identified differences between the two groups in terms of various variables and performed a multivariable analysis that incorporated these variables. The corresponding results are presented in Tables 1 and 3. In addition, we calculated the Mehran 2 risk score and performed a comparison between the two groups. This score was relatively high in our study compared to that in other studies because we only included patients with STEMI. We agree that incorporating pre- and post-procedural risk scores would aid in making decisions regarding patient risk prior to PCI. However, in our study, all patients belonged to a single group (patients with STEMI), and we were unable to gather data on the necessary variables, such as procedural bleeding and no flow, to calculate the risk score after the procedure. 

Patients with STEMI, the focus of our study, are at a high risk of developing PC-AKI. Interestingly, the PC-AKI group exhibited a higher risk score than did the group without PC-AKI. These findings suggest a significant correlation between the Mehran 2 risk scoring system and our study. We appreciate your valuable comment and acknowledge it as an important area for further research. We have added the following text to the Discussion:

“Incorporating both pre- and post-procedural Mehran 2 risk scores could be valuable in determining patient risk prior to PCI. However, our study only included a single group of patients with STEMI. Unfortunately, we were unable to collect data on the required variables, such as procedural bleeding and no flow, to calculate the risk score after the procedure. A large prospective study is required to validate the pre- and post-procedural Mehran 2 risk scores in patients undergoing PCI for STEMI [31].”

The median Mehran 2 risk score in the No PC-AKI group was 9 [IQR 8, 10], and in the PC-AKI group, it was 12 [IQR 11, 14]. There was a significant difference between the two groups (p < .001). We have added this information to Table 1.

 No PC-AKI AKI p-value

Mehran 2 risk score 9 (8, 10) 12 (11, 14) <.001

Data are presented as median (interquartile range).

Reference:

31.Guo Y, Xu X, Xue Y, Zhao C, Zhang X, Cai H. Mehran 2 contrast-associated acute kidney injury risk score: is it applicable to the Asian percutaneous coronary intervention population? Clin Appl Thromb Hemost 2022;28: 1–8.

7. The highest impact of CI-AKI on survival is noticeable at 30 days. Are data at 30 days available at least for a group of patients?

Response: Thank you for your comment. There were 59 patients with PC-AKI, among whom 43 were alive at 30 days (72.9%). However, there were 695 patients without PC-AKI, among whom 682 were alive at 30 days (98.1%).

We have added this information to Table 1.

8. In the discussion there should be more attention to the clinical significance of having a higher cut-off of 73 mL/min/1.73m2 that independently predicts the onset of CI-AKI for STEMI patients. What are the implications of classifying a wider population at risk of contrast nephropathy than the conventional threshold of 60 mL/min/1.73m2? How this new cut-off is actionable and what strategies should be implemented in the ED once patients at risk have been identified? 

Response: Thank you for your suggestion. We agree with your opinion. Firstly, at the beginning of the study, we consulted our statisticians regarding the statistical analysis. According to the central limit theorem, if the sample size (n) is large enough (typically greater than 30), the average will approximate a normal distribution. Accordingly, we performed the analysis using parametric methods. Our statisticians conducted the analysis in the same manner. We value your opinion on this matter. Based on your comment, we have conducted normality tests and revised Table 1 accordingly. Furthermore, in response to the Editor’s request, we have also revised to the demographic and test results. Based on the revised results, we reconducted the univariable and multivariable analyses. While our overall results remain unchanged, there have been some updates. The optimal eGFR cutoff value has changed to 61 mL/min/1.73 m2. 

It is widely known that an eGFR ≤60 mL/min/1.73 m2 is an independent risk factor for PC-AKI. Therefore, the increased incidence of PC-AKI at an eGFR ≤61 mL/min/1.73 m2 reported in this study is not surprising. However, when the eGFR cutoff level was calculated considering various risk factors, an eGFR ≤61 mL/min/1.73 m2 had increased clinical significance. It is important to note that the 2018 ESUR guidelines referenced in this study regard the eGFR level for PC-AKI risk to be <45 mL/min/1.73 m2 without considering various risk factors—this significantly differs from an eGFR ≤60 mL/min/1.73 m2. Therefore, considering various risk factors, it would be appropriate to focus on the result that an eGFR ≤61 mL/min/1.73 m2 was significantly associated with an increased risk of PC-AKI in this study. It is difficult to exclude all risk factors that contribute to PC-AKI and identify CI-AKI caused by contrast agents alone in a real-practice setting. The guidelines propose that the risk of PC-AKI is higher when the eGFR is <45 mL/min/1.73 m², without considering various risk factors after simple coronary angiography. However, until now, it has not been possible to determine an optimal cutoff for the development of PC-AKI, specifically in patients with STEMI, which is a more severe condition compared to other diseases. This study, being the first to suggest an optimal eGFR cutoff for predicting the occurrence of PC-AKI in patients with STEMI while considering various risk factors, is deemed highly original and holds significant clinical significance.

In addition, it can be very difficult to do so among patients with STEMI, as those without any risk factors for AKI are extremely rare. The main goal of the study was not to determine whether the use of contrast agents is the primary factor in PC-AKI among patients with STEMI. The significance of our findings is that caution and preventative measures should be taken when several risk factors for PC-AKI are present along with the use of contrast media, as PC-AKI can occur even if the eGFR level mentioned in the guidelines is higher. In particular, the cardiac function of patients with STEMI may be compromised, potentially resulting in limited fluid therapy. Considering the various risk factors in patients with STEMI, determining the optimal cutoff for the occurrence of PC-AKI can be beneficial in guiding fluid therapy as a preventive measure. Therefore, we believe that this information can be used as a guideline for fluid treatment to prevent AKI. Various preventive methods for PC-AKI, other than fluid therapy, are currently being studied. Adequate hydration is the primary approach for preventing PC-AKI, although additional measures such as high-dose statins (as indicated for secondary prevention), optimal hemodynamics, and discontinuation of nephrotoxic drugs, might offer significant benefits. Further large-scale prospective studies are needed to evaluate the effectiveness of these additional preventive measures.

9. Line 378, second sentence: correct the typo “Firs” with “First”.

Response: Thank you kindly for your comment. Accordingly, we have corrected the spelling mistake.

Response to Reviewer 2 comments

Reviewer #2: ACS is a high risk state due to highly thrombogenic state, increased inflammation, and decreased renal perfusion through vasoconstriction or hemodynamic instability. Where exposure of contrast medium can increase chances of CI AKI.

In the present paper, the authors present their data of optimal cut-offs of GFR for predicting CI-AKI. 

Response: We appreciate your kind comments and useful suggestions regarding our manuscript.

The introduction can be better rephrased. Following are some point regarding the same.

Comment 1: line 51: pPCI is a high-risk procedure for post-contrast acute kidney injury (PC-AKI). Here we can modify as - ACS is a high risk state due to highly thrombogenic state, increased inflammation, and decreased renal perfusion through vasoconstriction or hemodynamic instability. Where exposure of contrast medium can increase chances of CI AKI. Authors can give quotations of studies reporting such increased incidences of AKI in pPCI.

Response: Thank you for your suggestion. We have revised the Introduction according to your comment.

“However, acute coronary syndrome (ACS) is a high-risk condition owing to a highly thrombogenic state, increased inflammation, and decreased renal perfusion due to vasoconstriction or hemodynamic instability, where exposure to a contrast medium can increase the risk of post-contrast acute kidney injury (PC-AKI) [5-7].”

References

5.Narula A, Mehran R, Weisz G, Dangas GD, Yu J. Genereux P, et al. Contrast-induced acute kidney injury after primary percutaneous coronary intervention: results from the HORIZONS-AMI sub study. Eur Heart J. 2014; 35:1533-40.

6.Dong M, Jiao Z, Liu T, Guo F, Li G. Effect of administration route on the renal safety of contrast agents: a meta-analysis of randomized controlled trials. J Nephrol. 2012;25:290–301.

7.Solomon R. Contrast-induced acute kidney injury: is there a risk after intravenous contrast? Clin J Am Soc Nephrol. 2008;3:1242–1243.

Comment 2: line 56: Unlike CT- kindly use full form

Response: Thank you very much for your comment. Accordingly, we have revised “CT” to “computed tomography”.

Comment 3: for PCI, the medium: use contrast

Response: Thank you for your suggestion. We have revised “medium” to “contrast” accordingly.

Comment 4: ‘Although some reports suggest that hydration can reduce PC-AKI in patients with STEMI, to our knowledge, there are no effective therapeutic modalities for the prevention of PC-AKI.’ – Pre and post PCI hydration is the only intervention which has shown to reduce incidences of CI-AKI. Kindly give citiations too.

Response: We agree with your opinion and have revised the Introduction based on your recommendations. 

“The incidence of PC-AKI is high in patients undergoing cardiac procedures such as PCI [6,13]. Although some reports suggest that hydration can reduce PC-AKI in patients with STEMI, to our knowledge, there are no effective therapeutic modalities for the prevention of PC-AKI. Intravascular volume expansion can effectively prevent PC-AKI following invasive angiography. Numerous studies have been conducted to prevent PC-AKI, employing various strategies, such as appropriate risk stratification, reducing iodinated contrast dose, utilizing low or iso-osmolar contrast media, withholding nephrotoxic medications when necessary, and administering protective agents [14,15]. However, the use of contrast media is associated with multiple adverse effects; hence, it is a burden on physicians in the clinical setting [7].”

References

14.Michel P, Amione-Guerra J, Sheikh O, Jameson LC, Bansal S, Prasad A. Meta-analysis of intravascular volume expansion strategies to prevent contrast-associated acute kidney injury following invasive angiography. Catheter Cardiovasc Interv. 2021;98: 1120–1132.

15.Gupta RK, Bang TJ. Prevention of contrast-induced nephropathy (CIN) in interventional radiology practice. Semin Intervent Radiol. 2010;27: 348–359.

Comment 5: ‘Studies have shown that certain patient groups with baseline renal function, such as those with estimated glomerular filtration rates (eGFRs) <30, <45, or <60, are at an increased risk of developing PC-AKI’ – Kindly use 1 cutoff and their citation. Mostly <60 has been universally accepted as upper cut-off. 

Response: Thank you kindly for your comment. Accordingly, we have revised “<30, <45, or <60” to “<60” in the Introduction.

“Studies have shown that certain patient groups with baseline renal function, such as those with estimated glomerular filtration rates (eGFRs) <30, <45, or <60 mL/min/1.73 m2, are at an increased risk of developing PC-AKI.”

Comment 6: line 304: Hypotension : cardiogenic shock

Response: Thank you for your suggestion. We have revised “hypotension” to “cardiogenic shock” accordingly.

Comment 7: Line 324 ‘The fact that there was no significant difference in the contrast mediums amount between the group with PC-AKI and the group without PC-AKI in this study also supports these results’ – This conclusion is oversimplification of results. Studies have proved that the lower the contrast we use the less are the chances of developing CI-AKI and thus the advent of ultra-low contrast PCI in such high risk low GFR cases. There have been recently more robust data published which further strengthen the case for a ULC-PCI. PMID: 36007555. These data can be added to further enhance the evidence base for the same and improve the impact of the article. 

Response: Thank you for your valuable feedback. We have amended the Discussion based on your suggestion and the study by Shrivastava et al.

“The risk factors for developing PC-AKI in this study were similar to those reported in previous studies. Although there was no significant difference in the amount of contrast medium administered between the group with PC-AKI and the group without PC-AKI in our study, Shrivastava et al. demonstrated that the ultra-low contrast PCI protocol, compared to conventional PCI, is reasonably safe and effective in reducing the incidence of CI-AKI in a high-risk cohort [29]. Despite the significance of the type and amount of contrast medium administered in the development of PC-AKI, our results suggest that the etiology of AKI in patients with STEMI undergoing PCI is multifactorial, including the role of the contrast medium, rather than solely attributable to the contrast medium alone. This finding aligns with the results of a previous study [30].

Reference

29.Shrivastava A, Nath RK, Mahapatra HS, Pandit BN, Raj A, Sharma AK, et al. Ultra-low CONtraSt PCI vs conVEntional PCI in patients of ACS with increased risk of CI-AKI (CONSaVE-AKI). Indian Heart J. 2022;74: 363–368.

Comment 8: Addition of pre-procedural risk scores and post-procedure risk scores can be done which help objectively in decision making regarding the risk of the patient before PCI. The patients could have been classified according to their risk scores and would have given a more deeper insight on the causes of development of AKI in an individual.

Response: Thank you for your insightful opinion. We agree that incorporating pre- and post-procedural risk scores would aid in making decisions regarding patient risk prior to PCI. We apologize for any confusion caused. At the beginning of the study, we consulted our statisticians regarding the statistical analysis. According to the central limit theorem, if the sample size (n) is large enough (typically greater than 30), the average will approximate a normal distribution. Accordingly, we performed the analysis using parametric methods. Our statisticians conducted the analysis in the same manner. However, in response to the Editor’s request, we conducted normality tests and revised the demographic and test results accordingly.

Furthermore, we held discussions with our statisticians and proceeded to conduct a revised statistical analysis, incorporating further significant clinical variables mentioned by Reviewer 1. As a result, we identified differences between the two groups in terms of various variables and performed a multivariable analysis that incorporated these variables. The corresponding results are presented in Tables 1 and 3. In addition, we calculated the Mehran 2 risk score and performed a comparison between the two groups. This score was relatively high in our study compared to that in other studies because we only included patients with STEMI. We agree that incorporating pre- and post-procedural risk scores would aid in making decisions regarding patient risk prior to PCI. However, in our study, all patients belonged to a single group (patients with STEMI), and we were unable to gather data on the necessary variables, such as procedural bleeding and no flow, to calculate the risk score after the procedure. Patients with STEMI, the focus of our study, are at a high risk of developing PC-AKI. Interestingly, the PC-AKI group exhibited a higher risk score than did the group without PC-AKI. These findings suggest a significant correlation between the Mehran 2 risk scoring system and our study. We appreciate your valuable comment and acknowledge it as an important area for further research. 

The median Mehran 2 risk score in the No PC-AKI group was 9 [IQR 8, 10], and in the PC-AKI group, it was 12 [IQR 11, 14]. There was a significant difference between the two groups (p <.001). We have added this information to Table 1.

 No PC-AKI AKI p-value

Mehran 2 risk score 9 (8, 10) 12 (11, 14) <.001

Data are presented as median (interquartile range).

We have also amended the Discussion as follows:

“Incorporating both pre- and post-procedural Mehran 2 risk scores could be valuable in determining patient risk prior to PCI. However, our study only included a single group of patients with STEMI. Unfortunately, we were unable to collect data on the required variables, such as procedural bleeding and no flow, to calculate the risk score after the procedure. A large prospective study is required to validate the pre- and post-procedural Mehran 2 risk scores in patients undergoing PCI for STEMI [31]”

Reference:

31.Guo Y, Xu X, Xue Y, Zhao C, Zhang X, Cai H. Mehran 2 contrast-associated acute kidney injury risk score: is it applicable to the Asian percutaneous coronary intervention population? Clin Appl Thromb Hemost 2022;28: 1–8.

Comment 9: The average contrast consumption in the setting of pPCI was more than 100 ml which is acceptable in a pPCI setting. However what do the authors conclude in a setting of a low GFR what should the interventionalist do. Should he give less contrast or give more hydration? This articles conclusion seems there is no association with the amount of contrast use so such an exercise would prove futile. Kindly elaborate in discussion as to what this article and these cut-offs mean in real world.

Response: At the beginning of the study, we consulted our statisticians regarding the statistical analysis. According to the central limit theorem, if the sample size (n) is large enough (typically greater than 30), the average will approximate a normal distribution. Accordingly, we performed the analysis using parametric methods. Our statisticians conducted the analysis in the same manner. However, in response to the Editor’s request, we conducted normality tests and revised the demographic and test results accordingly. Based on the revised results, we reconducted univariable and multivariable analyses. While our overall results remain unchanged, there have been some updates. The optimal eGFR cutoff value has changed to 61 mL/min/1.73 m2. 

The risk factors for developing PC-AKI in this study were similar to those reported in previous studies. Although there was no significant difference in the amount of contrast medium administered between the group with PC-AKI and the group without PC-AKI in our study, Shrivastava et al. demonstrated that the ultra-low contrast PCI protocol, compared to conventional PCI, is reasonably safe and effective in reducing the incidence of CI-AKI in a high-risk cohort [29]. Despite the significance of the type and amount of contrast medium administered in the development of PC-AKI, our results suggest that the etiology of AKI in patients with STEMI undergoing PCI is multifactorial, including the role of the contrast medium, rather than solely attributable to the contrast medium alone. In particular, the cardiac function of patients with STEMI may be compromised, leading to limited fluid therapy. Considering the various risk factors in patients with STEMI, determining the optimal cutoff for the occurrence of PC-AKI can be beneficial in guiding fluid therapy as a preventive measure. Therefore, we believed that this information can be utilized as a guideline for fluid treatment to prevent AKI.

Reference

29.Shrivastava A, Nath RK, Mahapatra HS, Pandit BN, Raj A, Sharma AK, et al. Ultra-low CONtraSt PCI vs conVEntional PCI in patients of ACS with increased risk of CI-AKI (CONSaVE-AKI). Indian Heart J. 2022;74: 363–368.

Comment 10: Also what was the follow-up GFR of patients post intervention. Does primary PCI help the GFR in the longer run. A longer follow-up would have helped. As reperfusion helps improve cardiac hemodynamics and renal hemodynamics in the patients. So is pPCI benificial to the patient renal wise . what are authors opinions regarding it. 

Response: Thank you kindly for your comment. This study collected data solely on serum creatinine (Cr) and eGFR up to 3 days after PCI, using contrast as the diagnostic criterion for PC-AKI. Given that the diagnostic criteria rely on Cr values, we did not include eGFR results. As you correctly pointed out, we agree that PCI has the potential to improve both cardiac and renal hemodynamics, thereby aiding renal function recovery. The application of PCI may have a positive impact on preventing PC-AKI occurrence. However, due to the absence of long-term data, it is not possible to provide an opinion on the enduring effects of PCI on renal function within the scope of this study. We appreciate your valuable insights and consider them important for further research. Consequently, we have incorporated this point into the revised Limitations.

“Finally, this study focused on collecting data on serum creatinine (Cr) and eGFR within a 3-day period following PCI, with contrast serving as the diagnostic criterion for PC-AKI. PCI has the potential to enhance both cardiac and renal hemodynamics, which may contribute to the recovery of renal function. Additionally, PCI can play a beneficial role in preventing the occurrence of PC-AKI. However, to further validate the impact of PCI on renal function improvement through hemodynamic recovery and the prevention of PC-AKI development, it is essential to conduct large-scale prospective studies over a longer duration.”

Comment 11: what are authors views after the study as to which patients should receive ACEi and ARNis and on what day after intervention based on their data. Any expert comments as to minimize the risk of AKI and also utilize them for their benefits

Response: Thank you very much for your comment. In response to the Editor’s request, we conducted normality tests and subsequently revised the demographic and test results accordingly. Based on the revised results, we reconducted the univariable and multivariable analyses. The variable “after intervention day”" was based on the history of medication intake within 3 days after the procedure, specifically after ACEi/ARB administration. However, in the multivariable analysis, we did not find any statistically significant association between ACEi/ARB use after PCI and the development of PC-AKI.

---

## [Decision Letter · Decision Letter 1]

4 Sep 2023

PONE-D-23-10724R1Baseline eGFR cutoff for increased risk of post-contrast acute kidney injury in patients undergoing percutaneous coronary intervention for ST-elevation myocardial infarction in the emergency departmentPLOS ONE

Dear Dr. Beom,

Thank you for submitting your manuscript to PLOS ONE. After careful consideration, we feel that it has merit but does not fully meet PLOS ONE’s publication criteria as it currently stands. Therefore, we invite you to submit a revised version of the manuscript that addresses the points raised during the review process.

We look forward to receiving your revised manuscript.

Kind regards,

Satoshi Higuchi

Academic Editor

PLOS ONE

Journal Requirements:

**Additional Editor Comments:**

Dear authors,

Thank you for your kind response and effort on the review process. Your manuscript has improved significantly. However, I have only one concern. The original data was acquired prospectively; therefore, written informed consent should be obtained. Please clarify whether the investigators have obtain consent from patients.

Reviewers' comments:

Reviewer's Responses to Questions

**Comments to the Author**

1. If the authors have adequately addressed your comments raised in a previous round of review and you feel that this manuscript is now acceptable for publication, you may indicate that here to bypass the “Comments to the Author” section, enter your conflict of interest statement in the “Confidential to Editor” section, and submit your "Accept" recommendation.

Reviewer #1: All comments have been addressed

Reviewer #2: All comments have been addressed

2. Is the manuscript technically sound, and do the data support the conclusions?

Reviewer #1: Yes

Reviewer #2: Yes

3. Has the statistical analysis been performed appropriately and rigorously? 

Reviewer #1: Yes

Reviewer #2: Yes

4. Have the authors made all data underlying the findings in their manuscript fully available?

Reviewer #1: Yes

Reviewer #2: Yes

5. Is the manuscript presented in an intelligible fashion and written in standard English?

Reviewer #1: Yes

Reviewer #2: Yes

6. Review Comments to the Author

Reviewer #1: The authors have completed a thoroughly review of their paper addressing almost all reviewers’ comments.

The paper has greatly improved now but there are still few minor issues to be addressed.

1. Door-to-balloons time are now presented as median and IQR and the authors state that there are no significant difference with a p-value of 0.45. Yet in table 1 the reported p-value is <0.001. Please re-check the variable and the gaussianity test to confirm the non-normal distribution and present it accordingly.

2. In table 1, in the variable description along with the measurement units should be reported inside brackets whether a percentage or IQR or SD are reported. The same goes for header of table 2A.

3. Line 131 door-to-balloon times are still reported as mean and SD. Please provide a uniform reporting according to comment #1.

4. Line 313: a very simplified definition of cardiogenic shock is reported. I think the authors want to stress out the importance of hypoperfusion as a prominent risk factor for AKI. I suggest to remove the definition or use a more accurate one (SBP<90 mmHg for ≥30 min or use of pharmacological and/or mechanical support to maintain an SBP≥90 mmHg paired with evidence of end‐organ hypoperfusion which typically includes urine output<30 mL/h, cool extremities, altered mental status, and/or serum lactate >2.0 mmol/L.

5. Line 322: the word “impaired” is missing from the sentence “ACEi in patients with … glomerular filtration”

6. Line 324: replace “renal arterial disease” with “renal artery disease”

7. Line 370: the sentence has been truncated. Please add “factors for PC-AKI” after “several high-risk”

8. The final result of the multivariable analysis showed that eGFR and impaired cardiac output are the most important risk factors for AKI development in STEMI patients, regardless of previous medical history, medications and laboratory values. This finding is extremely important and should be emphasized. Following this concept a more detailed description of the hemodinamic status should be provided. It would be very interesting to compare between AKI and non-AKI patients the values of SBP, MBP, DBP, HR and if available also lactate levels. Those values can be measured at baseline, during or after the procedure and their significance can inform clinicians and interventional cardiologist about the proper management of the patient.

9. Please provide the reasons for IABP implantation (high risk anatomy, impending shock and so on).

Despite those minor comments I think the work of Dr. You and colleagues is extremely valuable and they should be praised for providing such important information for a clinical condition still profoundly neglected.

Reviewer #2: I think the authors have adequately addressed ALL QUERIES FROM MY SIDE. the authors have delved in hard to provide in-depth insight into the subject

7. PLOS authors have the option to publish the peer review history of their article (what does this mean?). If published, this will include your full peer review and any attached files.

Reviewer #1: No

Reviewer #2: **Yes: **Abhinav Shrivastava

---

## [Author Response · Author response to Decision Letter 1]

24 Sep 2023

We responded to the reviewer's comments and edited the text.

---

## [Decision Letter · Decision Letter 2]

17 Oct 2023

Baseline eGFR cutoff for increased risk of post-contrast acute kidney injury in patients undergoing percutaneous coronary intervention for ST-elevation myocardial infarction in the emergency department

PONE-D-23-10724R2

Dear Dr. Beom,

We’re pleased to inform you that your manuscript has been judged scientifically suitable for publication and will be formally accepted for publication once it meets all outstanding technical requirements.

Kind regards,

Satoshi Higuchi

Academic Editor

PLOS ONE

Additional Editor Comments (optional):

Reviewers' comments:

Reviewer's Responses to Questions

**Comments to the Author**

1. If the authors have adequately addressed your comments raised in a previous round of review and you feel that this manuscript is now acceptable for publication, you may indicate that here to bypass the “Comments to the Author” section, enter your conflict of interest statement in the “Confidential to Editor” section, and submit your "Accept" recommendation.

Reviewer #1: All comments have been addressed

2. Is the manuscript technically sound, and do the data support the conclusions?

Reviewer #1: Yes

3. Has the statistical analysis been performed appropriately and rigorously? 

Reviewer #1: Yes

4. Have the authors made all data underlying the findings in their manuscript fully available?

Reviewer #1: Yes

5. Is the manuscript presented in an intelligible fashion and written in standard English?

Reviewer #1: Yes

6. Review Comments to the Author

Reviewer #1: (No Response)

7. PLOS authors have the option to publish the peer review history of their article (what does this mean?). If published, this will include your full peer review and any attached files.

Reviewer #1: No

---

## [Editor Report · Acceptance letter]

19 Oct 2023

PONE-D-23-10724R2 

Baseline eGFR cutoff for increased risk of post-contrast acute kidney injury in patients undergoing percutaneous coronary intervention for ST-elevation myocardial infarction in the emergency department 

Dear Dr. Beom:

I'm pleased to inform you that your manuscript has been deemed suitable for publication in PLOS ONE. Congratulations! Your manuscript is now with our production department. 

Kind regards, 

on behalf of

Dr. Satoshi Higuchi 

Academic Editor

PLOS ONE